# Dip Hopping Technique and Yeast Biotransformations in Craft Beer Productions

Paolo Passaghe *, Lara Tat, Alba Goi, Luca Vit and Stefano Buiatti 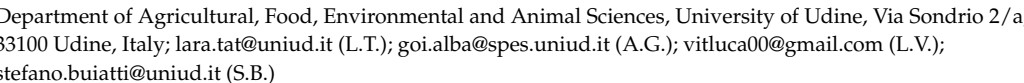

Department of Agricultural, Food, Environmental and Animal Sciences, University of Udine, Via Sondrio 2/a, 33100 Udine, Italy; lara.tat@uniud.it (L.T.); goi.alba@spes.uniud.it (A.G.); vitluca00@gmail.com (L.V.); stefano.buiatti@uniud.it (S.B.)

* Correspondence: paolo.passaghe@uniud.it; Tel.: +39-3462365112

**Abstract:** This paper evaluates the effects of an alternative hopping technique, called dip hopping, on beer. This technique involves infusing hops in hot water (or in a portion of wort) and subsequently combining the infusion with the wort (after wort cooling) directly in the fermenter when the yeast is added for fermentation. The reference beers were produced employing the "traditional" late hopping technique, and the experimental beers were produced using the dip hopping technique. A variety of hops with a significant concentration of essential oil and a strain of yeast with high β-glucosidic activity capable of releasing aromatic molecules from precursors supplied by hops were used. The samples were analysed in terms of alcohol content, degree of attenuation, colour, and bitterness. Sensory analysis and gas chromatography analysis were also performed. The data showed statistically significant differences between the reference beers and the experimental beers, with the latter featuring greater hints of citrus, fruity, floral, and spicy aromas. As an overall effect, there was an increase in the olfactory and gustatory pleasantness of the beers produced with the dip hopping technique.

**Keywords:** biotransformations; aroma; yeasts; fermentation; hop

## 1. Introduction

The hop aroma is difficult to characterise due to the complex composition of hop essential oils, the chemical and biological losses and transformations that constituents potentially undergo during the fermentation process, in addition to the additive and synergistic interactions between countless volatile compounds [1–7]. Yeast strains can affect aroma by chemically interacting with specific hop-derived compounds. Such reactions are commonly referred to as biotransformations [8,9]. It is known that certain strains of yeast have higher levels of enzymatic activity associated with biotransformation. Enzymatic hydrolysis occurs mainly thanks to 1,4-β-glucosidase (but also thanks to β-lyase), present in certain yeast strains used in fermentation [10–16]. Biotransformation of hop compounds by yeast strains is not limited to glycosides [17], since it is also expressed in monoterpenes (leaving the most complex and oxygenated terpenes unchanged). In fact, it has been observed that yeast is able to transform geraniol mainly into citronellol but also into linalool and nerol. The latter two are further converted into α-terpineol [18]. Hops can be used in accordance with various techniques. Traditionally, hops are added one or more times during the boiling process (kettle hopping). Boiling permits the conversion of α-acids into iso-α-acids, which is why hops with greater bittering compounds tend to be used [19–22]. A second option is the late hopping technique, in which hopping is carried out at the end of the boil, limiting the conversion of α-acids while favouring the retention of volatile molecules. Accordingly, the bittering contribution is limited, and varieties delivering a more significant essential oil content than bitter varieties tend to be used. Late hopping also increases the oxygenated portion of the aromatic compounds, contributing to floral and spicy aromas [23]. However, a significant proportion of essential oils are still

lost by means of evaporation, even if the hops are added late in the boiling process [24]. To limit the loss of aromatic molecules, a third technique used is dry hopping, which involves adding hops directly into the fermentation tank at cold temperatures. This favours the presence of volatile compounds that are more representative of the essential oils originally present in hops [25–30]. Nevertheless, it is commonly accepted that the technique leads to greater extraction of polyphenols, which, in combination with iso-α-acids, results in a bitterness described as harsh and medicinal [31–33]. Finally, it should be considered that, in terms of water and raw material use, the dry hopping process is particularly costly since a considerable amount of beer is lost when the hops are removed [34]. Lastly, a fourth hopping technique, called dip hopping, exists. This method involves steeping hop pellets in hot water or in a minimal quantity of hot wort. The mixture is then added directly into the fermenter at the same time as the yeast. This technique may be regarded as a halfway point between late and dry hopping, but there is very limited literature on the subject. At the EBC congress in Luxembourg in May 2013, researchers from the Japanese group Kirin (Kirin Holdings) presented this novel technique involving hops [35], which was then specified in a patent registered in the same year by Kirin Brewery Company Limited. The same group showcased new research at the Brewing Summit in San Diego [36]. Various professionals and hobbyists have experimented with the technique, yet there is scarce independent scientific research demonstrating the potential advantages or disadvantages of it. The experimental tests were designed to evaluate the true aromatic effect of the addition of hops employing the dip hopping technique compared to a "traditional" hopping technique. The tests conducted in this research aimed to improve the regulation of the beer's olfactory profile, both in terms of its composition and persistence over time. The findings obtained can also contribute to the enrichment of traditional hops, expanding the range of flavours that the latter can impart to beer. Furthermore, the purpose of these experiments is to offer practical insights for brewing beers with an increasingly sought-after flavour.

## 2. Materials and Methods

### 2.1. Experimental Design

During this research, six craft beers were produced using a particularly flavouring hop (Idaho) and a selected yeast with a high β-glucosidase activity, while comparing the traditional late hopping technique with the new dip hopping technique. Analyses were conducted on the beers to determine the alcohol content, degree of attenuation, color, and bitterness. Additionally, sensory analysis and gas chromatography analysis were performed (Figure 1).

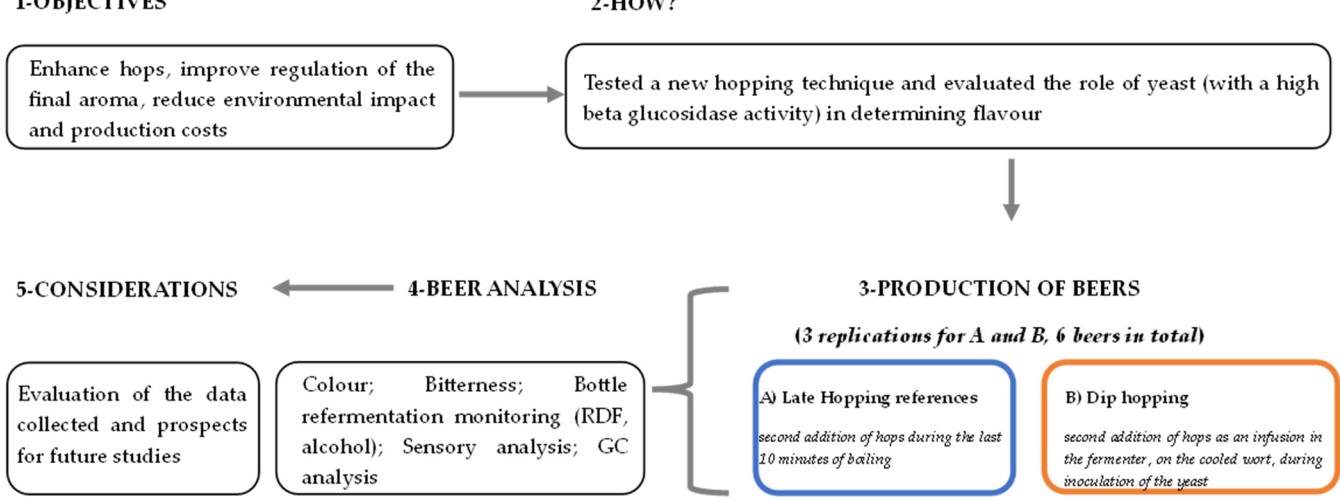

**Figure 1.** Experimental design.

### 2.2. Raw Materials

T90 Idaho 7 pellet hops, harvested in 2021, with an α-acid content of 12.3% and 2 mL of total oil per 100 g of hops (produced by Yachima Chief Hops LLC, 306 Division St., Yakima, Washington, DC, USA) were used. The Finest Pale Ale Golden Promise (Simpsons Malt, Castleford, West Yorkshire, UK) was used as a malt. A yeast that contributed only in an insignificant manner to the aroma, which had a notable glycosidic activity, was used (packaged in vacuum bags of 11 g): LalBrew Bry-97 (American West Coast Ale Yeast), manufactured for Danstar Ferment AG by Lallemand GmbH, Ottakringwestrasse 89, Vienna, Austria. All raw materials were obtained from P.A.B. SRL, Mr. Malt®, Pasian di Prato, Udine, Italy.

### 2.3. Beer Production

Two types of beer—a reference beer (late hopping) and an experimental beer (dip hopping)—were produced in the India Pale Ale style (IPA, Table 1). India Pale Ale is a style of beer originally brewed in the late 18th century in the British Empire. These beers had a higher hop content and higher alcohol levels than a typical light beer, so they could endure the lengthy journey to the colonies. A broad array of beer styles make up the modern IPA family, all of which are distinguished by an intense hop flavour [37]. Late hopping (traditional technique, A beers) and dip hopping (novel technique, B beers) techniques were compared using a single variety of flavouring hops (Idaho).

**Table 1.** Technical data sheet of the beer produced.

| Style | India Pale Ale |
|---|---|
| Original Gravity (OG) [a] | 1052 |
| °Plato [b] | 13 |
| Final litres (post-boiling) | 20 L |
| Bitterness (IBU) | 34 |
| Colour (EBC) | 9 |
| Alcohol content% | 5.5% |
| Mash notes | Mashing: 60 min at 65 °C<br>Mash-out: 10 min at 77 °C |
| Fermentation notes | Primary: 14 days at 20 °C |

[a] the density of the wort at standard temperature and pressure. [b] g of extract for every 100 g of wort.

Three replicas of each technique were carried out, which resulted in a total of six craft beers (Table 2). The production order was randomised using Excel. The water used to produce the beers had the following characteristics: Ca 63 mg/L, Mg 15 mg/L, Na 2 mg/L, Cl 3 mg/L, and $SO_4$ 15 mg/L [38]. A total of 4.4 kg of malt were used for both productions. A total of 23 L of water were placed in a Braumeister (Speidel, Ofterdingen, Germany), which were heated up to 65 °C. The water was acidified with 8.8 mL of lactic acid (80% *v/v*, mash pH 5.6), and the following salts were added: $CaCl_2$ (2.25 g), NaCl (1.62 g), and $CaSO_4$ (3.32 g). Upon reaching the desired temperature, the ground malt was added, and mashing began, which lasted 60 min. After this time, the mash-out process began: the temperature was raised to 77 °C, and once it was reached, the process was paused for 10 min (Table 1). At the end of the pause, manual sparging was carried out with water at 77 °C (2.6 L) that had lactic acid added (0.84 mL, 80% *v/v*, sparge pH 5.8) and salts: $CaCl_2$ (0.25 g), NaCl (0.18 g), and $CaSO_4$ (0.38 g). After 20 min of sparging, the wort was heated to 100 °C and kept at this temperature for 60 min (boiling/cooking process).

**Table 2.** Quantities of raw materials and codes of beers.

| Raw Materials | Quantity | Beer Code |
|---|---|---|
| Malt: Simpsons Pale Ale Golden Promise | 4400 g | A, B |
| Hop: Idaho 7. 60 min boiling | 13 g | A, B |
| Hop: Idaho 7. 10 min boiling (late hopping) | 82 g | A |
| Hop: Idaho 7. In 1.5 L of water at 77 °C for 30 min (dip hopping) | 88 g | B |
| Yeast: Dry–LalBrew Bry-97 | 22 g | A, B |

The reference provided for the addition of 13 g of hops at the beginning of boiling and another 82 g employing the late hopping technique 10 min before the end of the boiling stage (during manual whirlpool, Table 2). Cooling to room temperature followed. The experimental beers were produced by first adding 13 g of hops at the beginning of boiling and then weighing 88 g of hops, which were added in 1.5 L of hot water (77 °C) and infused for 30 min (Table 2). Subsequently, the infusion was added to the cooled wort (15–17 L). After which, 1.5 L of water was added to the cooled worts of the reference beers (A beers) to standardise them with the experimental beers (B beers). The amount of hops used for dip hopping was therefore slightly higher (88 g) than that used for late hopping (82 g). This decision was made to limit the difference between A and B beers in terms of perceived bitterness, making a sensory comparison possible. In fact, the hops added during the last 10 min of boiling (late hopping) had a higher yield of isomerization of $\alpha$-acids into iso-$\alpha$-acids (bitter) compared to the infusion of hops for 30 min in 1.5 L of water at 77 °C [39]. This factor should be considered since the perception of bitterness influences the aromatic profile (olfactory and retro-olfactory) of the finished beer through the cross-modal perception "mechanism" [40]. For both types of beer, when the wort reached room temperature (20 °C), the yeast was added (two sachets totaling 22 g) and fermented (primary and secondary fermentation) for 14 days (high fermentation, as per Tables 1 and 2). Following a 7-day dip hopping process and 14 days of fermentation employing the late hopping technique, the beer was transferred into another 20-L plastic fermenter supplied by the Italian company Mr. Malt® (Pasian di Prato, Udine, Italy). This facilitated the elimination of any compounds and yeast cells that had settled at the bottom. On day 14, a glucose monohydrate solution was added to the second container of both beers (A and B) to aid bottle conditioning. The quantity was calculated so as to obtain 2.4 volumes of carbon dioxide (litres of carbon dioxide per litre of beer) after refermentation, based on the fact that 4 g of sucrose dissolved in 1 L of beer produce 1 volume of carbon dioxide. After this, the beer was poured into 0.5-L bottles. Bottle conditioning was carried out at room temperature.

*2.4. Beer Characterisation Analysis*

After 10 days of refermentation, beer colour analysis (spectrophotometric method) was performed in accordance with EBC (European Brewery Convention) method 9.6 [41]. After 10 days of refermentation, beer bitterness was analysed in accordance with EBC method 9.8 [41]. The determination of the main bittering compounds, i.e., iso-$\alpha$-acids, was performed by means of spectrophotometric measurement. Alcohol content (%*v/v*) and real degree of fermentation (RDF) were measured using the Alcolyzer Beer Analysing System (Anton Paar, Ostfildern, Germany) and the Anton Paar Density Meter (Anton Paar, Germany). In particular, the instrument comprises an Alcolyzer Plus module for measuring alcohol content and an Anton Paar density meter. Comparison studies of the results (alcohol percentage) obtained with this instrument and those determined with the official method (EBC method 9.2.1) [41] showed a non-statistically significant deviation between the mean values obtained with the two methods and an Anton Paar standard deviation of 0.025% *v/v* [42]. Determination was performed at zero and at ten, twenty, and thirty days after bottling.

2.4.1. Sensory Analysis and Statistical Data Processing

The various sensory analyses were carried out simultaneously, three months after bottling, in rooms set up in accordance with UNI-EN ISO 8589 standards concerning the sensory analysis of food products [43]. The sensory testing carried out employed a differentiation by attributes method with evaluation on a continuous scale [44]. The software Smart Sensory Box (Smart Sensory Solutions S.r.l., Sassari, Italy) was used to create the tasting plan and collect data. In the session, each judge tasted a total of six samples—the three reference replicates (A beers) and the three experimental replicates (B beers), which were proposed in a randomised and balanced order according to the schedule provided by the software. A panel of 16 judges carried out the testing. All judges (expert and non-expert consumers) were instructed on the meaning of the proposed attributes and the order of evaluation. Each attribute was evaluated using a continuous intensity line. Sensory scores were determined by measuring the distances of each point from the origin. The tasting card proposed to the judges and input in the software are shown in Table 3.

**Table 3.** Tasting card.

| | |
|---|---|
| Visual attributes | Foam |
| | Foam texture |
| | Colour |
| Olfactory attributes | Spicy |
| | Citrus |
| | Fruity |
| | Herbal |
| | Floral |
| | Garlic |
| | Other (please specify) |
| | Olfactory pleasantness |
| Defects | Medicinal |
| | Oxidised |
| | Reductive |
| Taste Attributes | Bitter |
| | Sweet |
| | Body |
| | Sparkle |
| Retro-olfactory attributes | Aroma richness |
| | Earthy notes |
| | Persistence |
| | Other (please specify) |
| | Pleasantness in the mouth (gustatory and retro-olfactory) |

The research adhered to the ethical guidelines set by the University of Udine, and all participants provided their informed consent before taking part in the study. The participants also confirmed that they did not have any known disorders triggered by gluten consumption.

In order to process the data, for each sensory analysis session, the test scores provided by the judges underwent normalisation [44], so as to eliminate the effects of a subjective use of the rating scales. Using the normalised data, a correlation analysis was made of the scores of each individual judge with the group average calculated, excluding the judge himself, descriptor by descriptor (Senstools for Windows, version 2.3). The average of the correlation coefficients on all attributes (averaged correlation coefficient) enabled the exclusion of the judges, who negatively correlated with the rest of the panel. After this elimination, the normalised data underwent statistical analysis (Statistics for Windows, version 8). Panelists with a positive correlation to descriptors close to significance were reintroduced for data processing, despite having a negative mean correlation coefficient, to assess whether a statistical difference could be attained. Preliminarily, the normality of the data (Kolmogorov–Smirnov test, Shapiro–Wilk W test) and the homogeneity of the

variances (Levene test, Bartlett test) were verified. For several descriptors, the conditions for carrying out the analysis of variance (ANOVA), i.e., normality and homoscedasticity of the data, were lacking. Therefore, processing continued with a non-parametric analysis. The Mann–Whitney U test (comparison test between two independent samples or groups: hopping techniques A and B), the Kruskal–Wallis test, and the median test associated with the multiple comparison of the average ranks for all groups (comparison test between several independent samples or groups: the three treatment replicates for the two hopping techniques) were used. Finally, the averages were calculated on the normalised data, and the same averages were used to analyse the main components (PCA).

2.4.2. Gas Chromatography Analysis and Statistical Data Processing

At the same time as the sensory analysis, the volatile components in the content of the three bottles for each production of the two hopping techniques (18 samples) were analysed. The volatile aromatic portion was determined by means of SPME-GC-MS in accordance with adaptation to the autosampler method developed by Tat et al. (2005) [45]. Certain variants were also used when preparing the sample for microextraction. The samples were placed in the refrigerator at 4 °C (to limit the formation of foam), and at the time of use, 10 mL of each sample was extracted under a nitrogen flow; rather than being extracted from the bottle with a pipette, the beer was poured into a 10 mL flask, and once the foam had reduced, the volume was adjusted and internal standards were added at a rate of 0.1 mL of a 96% mixed ethanol solution. The mixture consisted of 0.00913 g/L of $\gamma$-terpinene, 0.0958 g/L of 3-octanol, and 0.0852 g/L of ethyl dodecanoate. Once the sample solution was produced, it was transferred to the vials for instrumental analysis. A 2 cm triphasic fibre (Supelco®, Merck KGaA, Darmstadt, Germany) was used with a sampling temperature of 40 °C for 15 min (after 15 min of prior thermal balancing of the sample before microextraction). The samples were analysed using a gas chromatography system manufactured by Shimadzu, consisting of:

- an autosampler (HTA modello HT2800T, Brescia (BS), Italy);
- a gas chromatograph (GC 2030 Nexis Shimadzu Italia S.r.l., Milan, Italy) with a DB-WAX-MS column measuring 30 m by an internal diameter of 0.25 mm, with a 0.25-μm thick film;
- a mass spectrometer (GCMS-QP2020 NX Shimadzu Italia S.r.l., Milan, Italy) comprising an electronic impact source and a quadrupole analyser.

The following methods were applied:

- isotherm of 5 min at 40 °C;
- temperature increase from 40 °C to 250 °C with a 4 °C/minute ramp and a final isotherm of 15 min;
- injector at 250 °C and helium as a carrier gas (0.9 mL/minute flow);
- splitless-type injection for 3 min;
- transfer line at 240 °C;
- source at 200 °C.

The mass spectrometer operated in SCAN mode (scan range $m/z$ 25–400). Identification of volatile compounds was accomplished using the library of the data processing programme (NIST 08) and by calculating retention indices (Kovats indices) with injections of paraffins and comparing them with those of the bibliography [46]. Evaluation of the relative quantities (expressed as internal standard equivalents) was carried out by integrating the areas of the chromatographic peaks in total current or single $m/z$ ion when peaks overlapped. The peak areas of the compounds of interest were compared with those of the closest chemical internal standard ($\gamma$-terpinene for terpenes, ethyl dodecanoate for esters, 3-octanol for terpenols, and all other chemical classes; see Appendix A). The data were acquired and processed using the software GCMS Solution. Concentrations of the various compounds were used to perform a PCA analysis on the three bottles for each production of the two hopping techniques (18 samples).

The data collected with the SPME-GC-MS analysis were reprocessed through non-parametric statistical tests (since the conditions for parametric processing were not present in full) on the sum of the concentrations of compounds belonging to certain chemical classes (terpenes, terpenols, ketones, acids, alcohols, acetic esters, ethyl esters, and higher alcohol esters). The Kruskal-Wallis test and the median test associated with the multiple comparison of the average ranks for all groups (comparison test between several independent samples or groups: the three treatment replicates for the two hopping techniques) were used.

## 3. Results and Discussion

### 3.1. Colour and Bitterness

As can be noted in Table 4, higher values than the 9 EBC indicated in the sheet (Table 1) were collected for all beers (with the exception of samples A1 and A2), and the dip hopping technique appears to favour an increase in colour in the finished beer. The infusion conditions may actually favour the extraction of polyphenolic fractions [47]. Some researchers have found that the addition of polyphenols in a model system increases the perception of bitterness (with the same iso-$\alpha$-acids) and "fullness" (mouthfeel) [48–50]. Regarding bitterness, despite the use of a greater amount of hops in dip hopping, samples A (reference beers) have higher IBU values than the values obtained for the experimental beers (B beers) (Table 4).

**Table 4.** Bitterness and colour of the 6 beer samples, 10 days after bottling (data expressed as mean $\pm$ standard deviation; $n = 4$ for bitterness, $n = 3$ for colour). A = late hopping, B = dip hopping; the number refers to production replicas.

| Beer Code | Bitterness (IBU) | Colour (EBC) |
|:---:|:---:|:---:|
| A1 | $37 \pm 1$ | $9 \pm 0$ |
| A2 | $44 \pm 0$ | $9 \pm 0$ |
| A3 | $39 \pm 0$ | $10 \pm 0$ |
| B1 | $36 \pm 1$ | $11 \pm 0$ |
| B2 | $36 \pm 0$ | $11 \pm 0$ |
| B3 | $31 \pm 0$ | $12 \pm 0$ |

### 3.2. Alcohol Content and Real Degree of Fermentation

Ethanol and carbonation levels have been shown to affect the polarity and, hence, the retention or partitioning of many volatile compounds. Therefore, the level of carbonation and the alcohol content of the beers tested should also be checked as far as possible for sensory and gas chromatography analyses [51]. The alcohol content and degree of fermentation (RDF) analyses show the variability existing between the individual bottles (Table 5). RDF values (after thirty days of refermentation) range from a minimum of $69.40 \pm 0.42\%$ (beer A3) to a maximum of $71.29 \pm 0.18\%$ (beer B3), as can be seen in Table 5. All samples experienced an increase in alcohol content and RDF values while stored in bottles for one month. This increase is a consequence of the sugar added and oxygenation during the bottling process, which causes beer to ferment. Without exception, all samples contain alcohol quantities similar to the target values shown on the technical data sheet (Table 1). While the variation of the alcohol content in the first 10 days shows that the dip hopping technique does not affect the fermentation process. The comparable patterns between the reference beers (A) and the experimental beers (B) suggest, in fact, that the terpenes provided by the infusion, at these concentrations, do not have a negative effect on yeasts.

**Table 5.** Alcohol and RDF of the 6 beer samples at point zero (bottling) and after 10, 20, and 30 days (data expressed as mean ± standard deviation; *n* = 3). A = late hopping, B = dip hopping; the number refers to production replicas.

| Beer Code | Alcohol (% *v/v*) Refermentation Days | | | | RDF (%) Refermentation Days | | | |
| --- | --- | --- | --- | --- | --- | --- | --- | --- |
| | 0 | 10 | 20 | 30 | 0 | 10 | 20 | 30 |
| A1 | 5.68 ± 0.02 | 6.11 ± 0.00 | 6.08 ± 0.03 | 6.17 ± 0.01 | 65.45 ± 0.38 | 69.40 ± 0.98 | 69.02 ± 0.10 | 69.92 ± 0.56 |
| A2 | 5.51 ± 0.01 | 5.95 ± 0.17 | 6.14 ± 0.01 | 5.96 ± 0.02 | 67.66 ± 0.02 | 68.66 ± 0.58 | 70.38 ± 0.02 | 69.78 ± 0.10 |
| A3 | 5.39 ± 0.08 | 5.85 ± 0.02 | 6.06 ± 0.01 | 6.20 ± 0.01 | 64.03 ± 0.42 | 67.33 ± 0.05 | 68.50 ± 0.04 | 69.40 ± 0.42 |
| B1 | 5.66 ± 0.03 | 6.19 ± 0.00 | 6.26 ± 0.02 | 6.10 ± 0.02 | 65.12 ± 0.27 | 69.55 ± 0.00 | 70.29 ± 0.05 | 69.73 ± 0.04 |
| B2 | 5.51 ± 0.01 | 5.97 ± 0.01 | 6.04 ± 0.06 | 5.95 ± 0.02 | 64.67 ± 0.08 | 69.48 ± 0.02 | 69.95 ± 0.27 | 69.56 ± 0.07 |
| B3 | 5.56 ± 0.001 | 6.05 ± 0.12 | 6.06 ± 0.01 | 6.38 ± 0.02 | 67.81 ± 0.05 | 68.77 ± 1.09 | 71.11 ± 2.07 | 71.29 ± 0.18 |

### 3.3. Sensory Analysis

In the PCA analysis (Figure 2), the experimental B samples are clearly distinguished from the reference A samples. By examining the descriptors (Figure 3), a positive effect of dip hopping on B samples (fruity, citrus, spicy, floral, and olfactory pleasantness) is noted. In late hopping (A beers), on the other hand, more negative descriptors emerge (medicinal, oxidised, earthy).

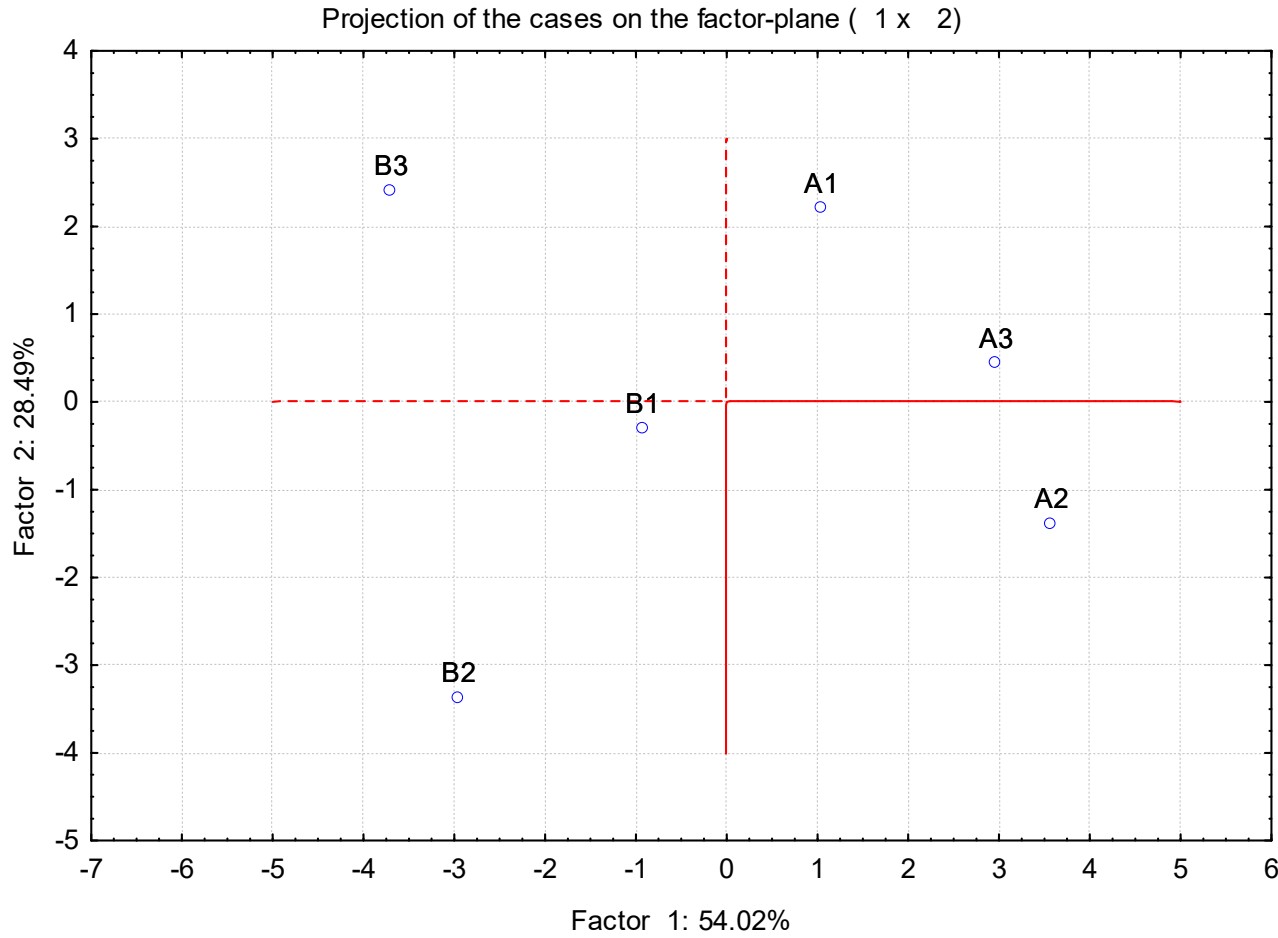

**Figure 2.** PCA of sensory analyses relating to late hopping (A) and dip hopping (B) 1, 2, and 3 are treatment replicates.

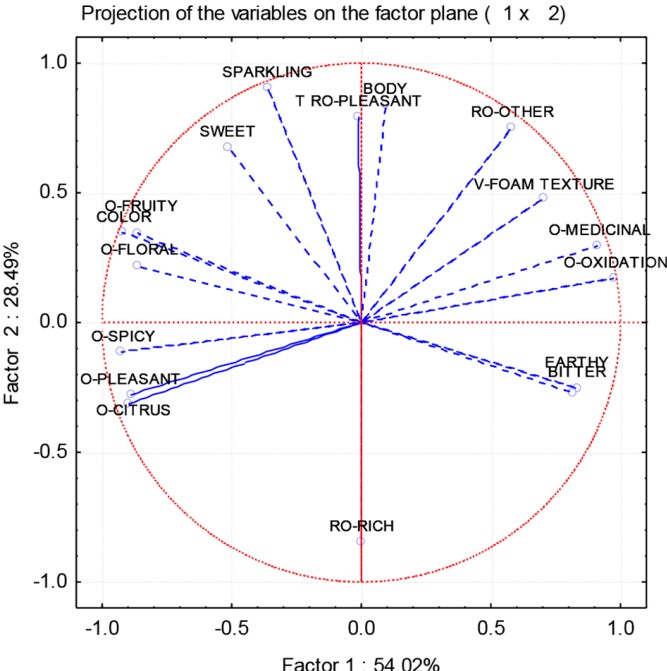

**Figure 3.** Distribution of PCA variables on sensory data. O = olfaction, RO = retro-olfaction; T = taste; V = visual.

Results of a non-parametric analysis of the descriptors (Mann–Whitney U test), with treatment replicates grouped together, showed numerous statistically significant differences (Figure 4): floral, citrus, and spicy aromas were significantly more present in the experimental beers (B), and olfactory pleasantness was also greater. The same descriptors also resulted in significant results in the non-parametric Kruskal–Wallis analysis and the median test, applied to all replicates (Figure 5).

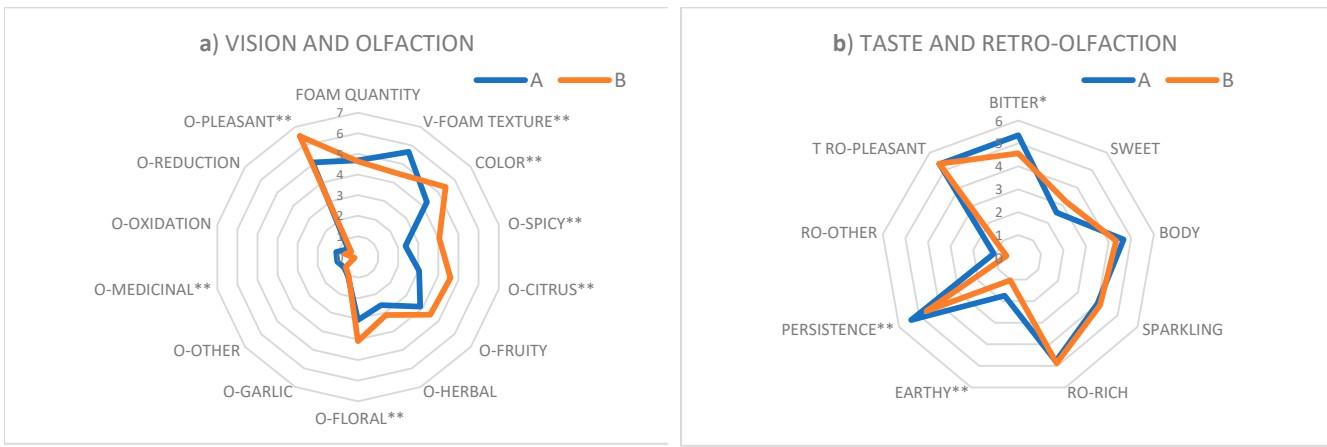

**Figure 4.** Representation of the normalised sensory data averages comparing the late-hopping beers (A) with the dip hopping beers (B) produced with Idaho (**a**), vision and olfaction; (**b**), taste and retro-olfaction). Statistically significant differences in the various attributes based on the Mann–Whitney test are highlighted with double asterisks (*p*-value ≤ 0.05). A single asterisk marks the reaching of significance subsequent to panellists with a positive correlation for specific attributes being reintroduced for data processing, despite having a negative mean correlation coefficient.

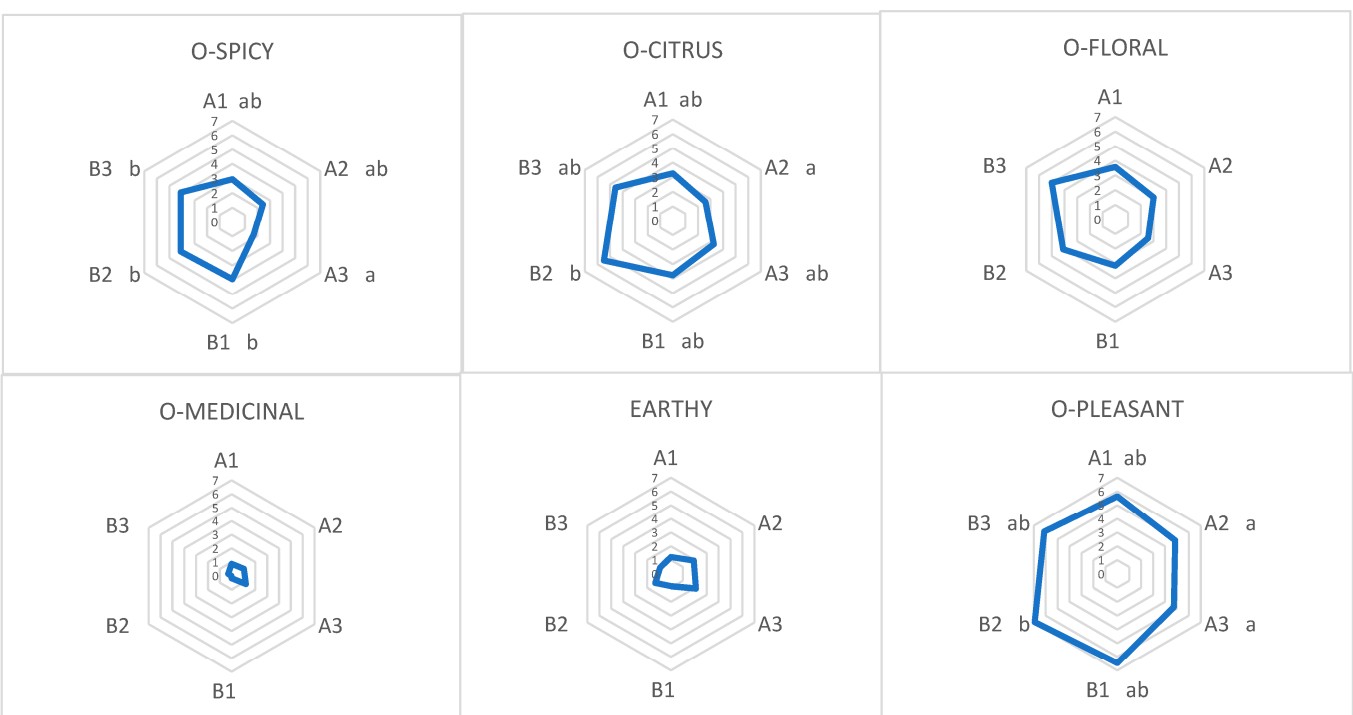

**Figure 5.** Representation of the normalised sensory data averages used in the Kruskal–Wallis tests and the median test, relating to late hopping beers (A) and dip hopping beers (B) produced with Idaho 1, 2, and 3 are treatment replicates. Statistically significant differences between the reference beers and the experimental beers related to specific attributes (*p*-value ≤ 0.05) based on the multiple comparison test of the mean ranks for all groups are represented with different letters.

*3.4. Gas Chromatography Analysis*

The differences observed in the sensory analysis (experimental B beers richer in fruity, spicy, and floral aromas) were confirmed by the chromatographic analysis (Figures 6 and 7), thus underscoring the effectiveness of dip hopping in extracting flavouring compounds imparted by hops. Indeed, it can be supposed that negative descriptors (earthy notes, medicinal, garlic, etc.) are perceived less as a result of a masking effect generated by the greater richness and aromatic complexity of the beers produced employing this technique. In particular, Figure 7 shows how terpenes, terpenols, alcohols, ketones, and higher alcohol esters—responsible for citrus and fruity aromas—are significant in the beers produced using dip hopping. Idaho is a variety of hops with essential oils particularly rich in the so-called "survivable compounds" (linalool, geraniol, methyl geraniate, 2 and 3-methylbutyl 2-methylpropanoate, and 2-nonanone).

These are compounds that remain in the finished product and significantly influence its aromatic characterization [52]. As shown in Figures 8–13, the experimental hopping technique accentuates the presence of these compounds. The concentration of β-myrcene (Figure 8) and the medium in which it is found can both affect the sensory attributes it imparts. β-myrcene has been combined with numerous aromatic notes, including lime [53], peppery, balsamic, plastic/synthetic [54], metallic, geranium-like [5], and spicy notes [55]. Kishimoto et al. (2005) [3] suggested that the resinous character attributable to β-myrcene is the result of the interaction of the same with other compounds (e.g., containing sulphur) present in traces. Sulfur-containing compounds are present only in trace or undetectable amounts, yet they are still essential in defining the olfactory profile of the product [25].

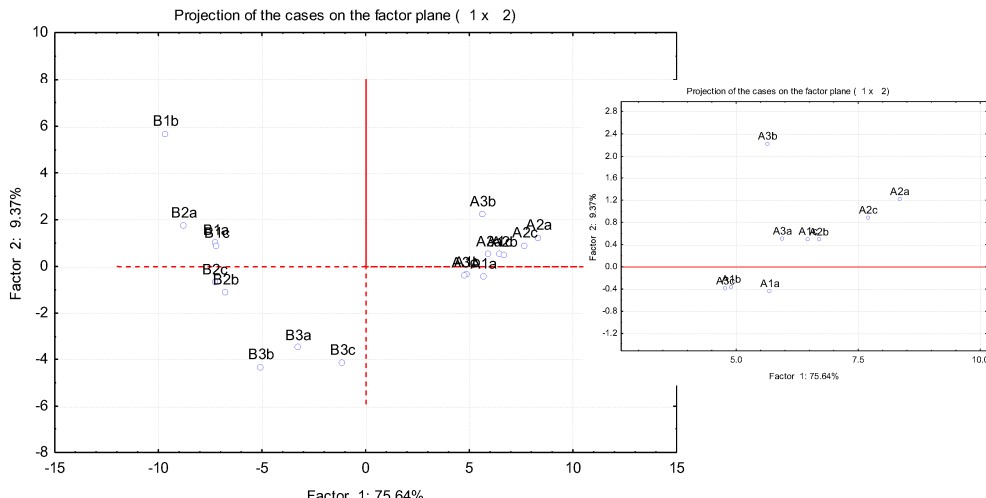

**Figure 6.** PCA analysis of gas chromatographic data relating to beers produced with late hopping (A) and dip hopping (B) techniques. 1, 2, 3 = treatment replicates; a, b, c = different bottles.

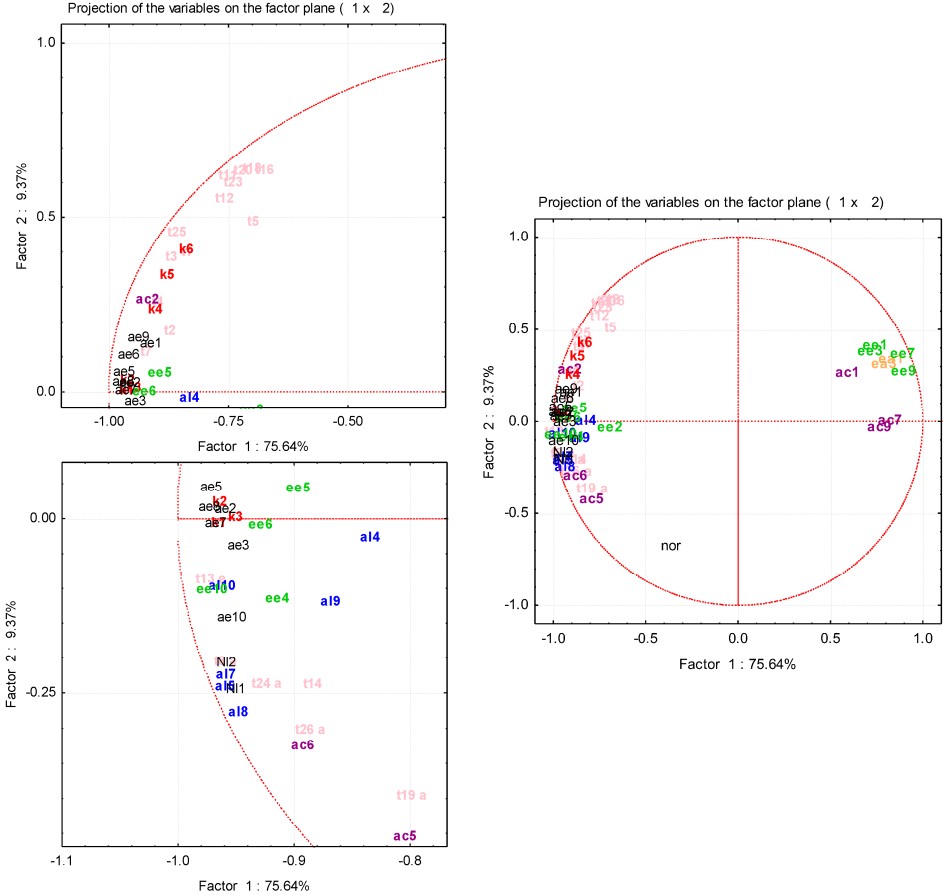

**Figure 7.** Distribution of the descriptors in accordance with the PCA analysis of the gas chromatographic data of the beers produced with the addition of Idaho employing late and dip hopping techniques. t = terpenes and terpenols (pink), k = ketones (red), al = alcohols (blue), ae = higher alcohol esters (black), ac = acids (purple), ee = ethyl esters (green), ea = acetic esters (yellow), nor = norisoprenoids (black), NI1 and NI2 = Not Identified. The number associated with the codes refers to the specific compounds listed in the Appendix A (Tables 1–3).

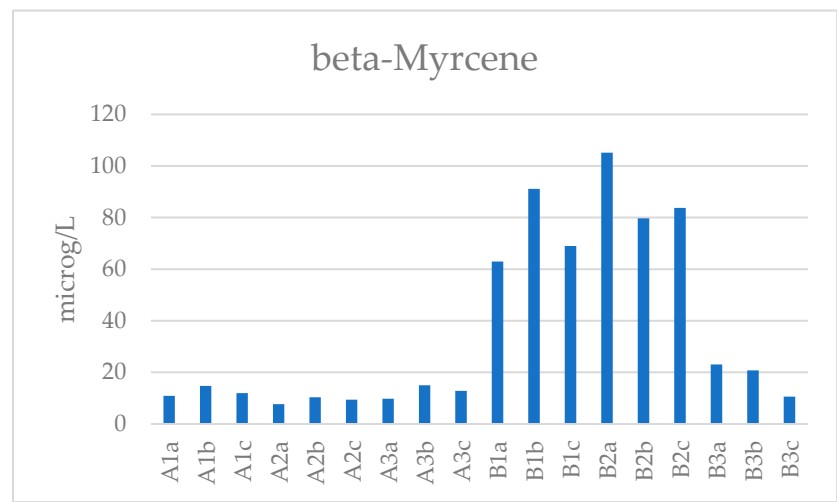

**Figure 8.** Concentration (SPME-GC-MS analysis) of β-myrcene in the late hopping (A) and dip hopping (B) beers. 1, 2, 3 = treatment replicates; a, b, c = different bottles.

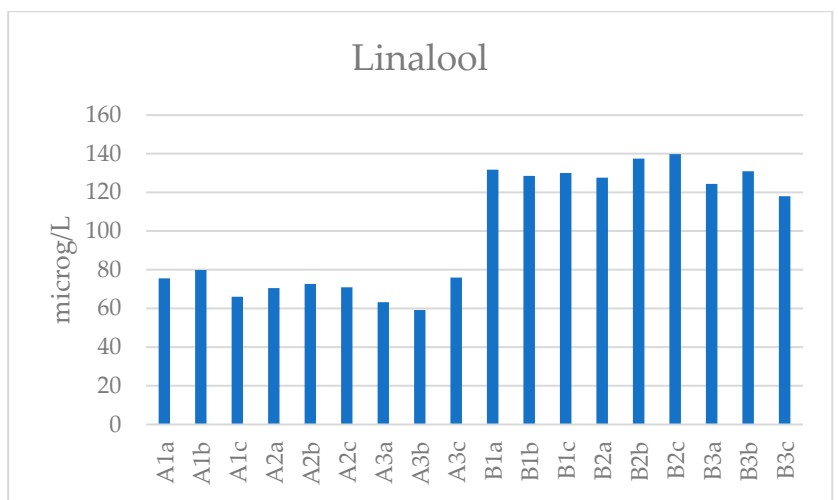

**Figure 9.** Concentration (SPME-GC-MS analysis) of linalool in the late hopping (A) and dip hopping (B) beers 1, 2, and 3 = treatment replicates; a, b, and c = different bottles.

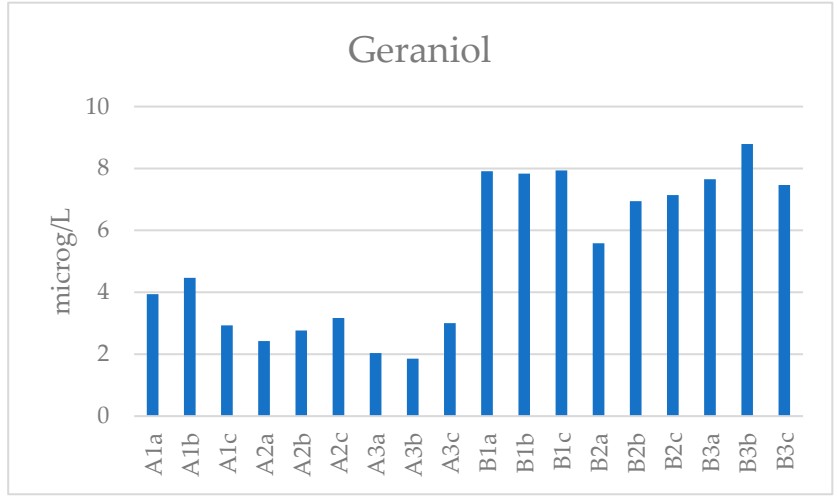

**Figure 10.** Concentration (SPME-GC-MS analysis) of geraniol in the late hopping (A) and dip hopping (B) beers. 1, 2, 3 = treatment replicates; a, b, c = different bottles.

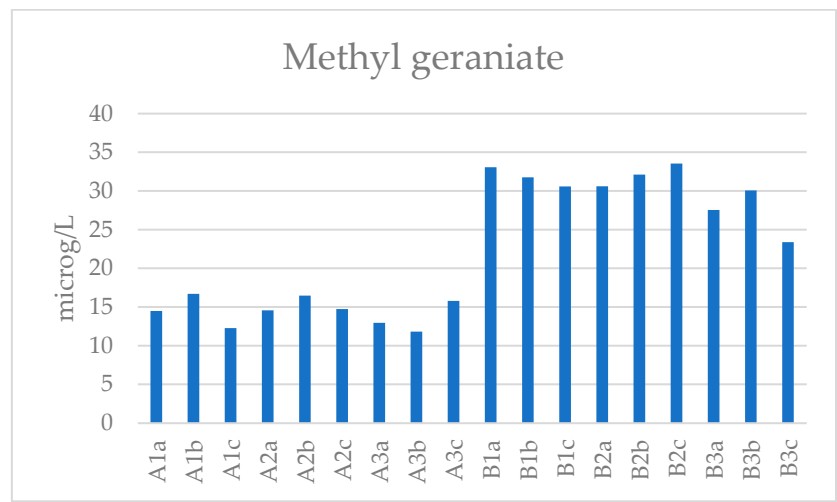

**Figure 11.** Concentration (SPME-GC-MS analysis) of methyl geraniate in the late hopping (A) and dip hopping (B) beers. 1, 2, 3 = treatment replicates; a, b, c = different bottles.

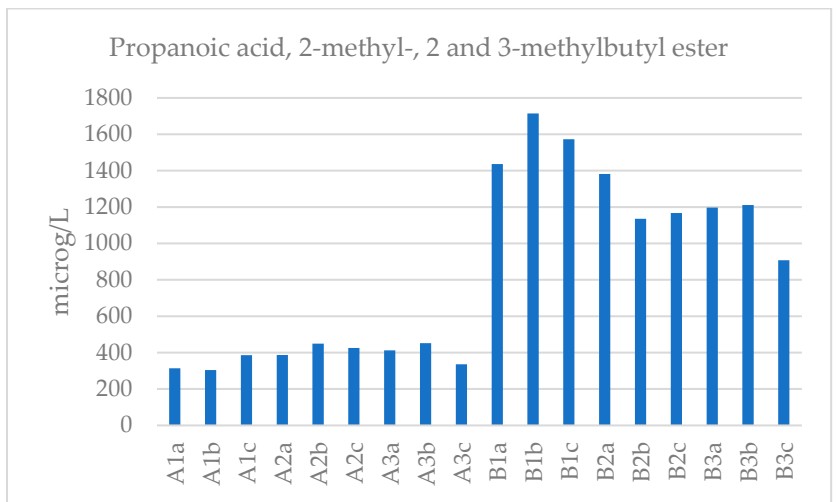

**Figure 12.** Concentration (SPME-GC-MS analysis) of 2 and 3-methylbutyl 2-methylpropanoate in the late hopping (A) and dip hopping (B) beers. 1, 2, 3 = treatment replicates; a, b, c = different bottles.

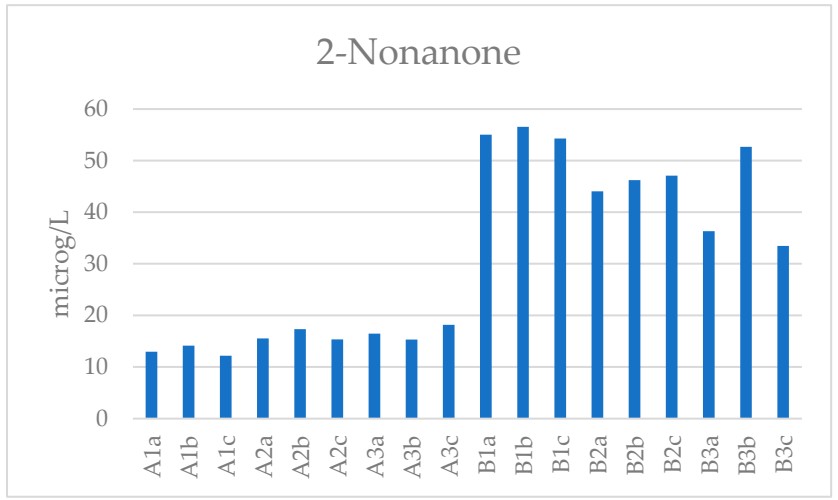

**Figure 13.** Concentration (SPME-GC-MS analysis) of nonanone in the late hopping (A) and dip hopping (B) beers. 1, 2, 3 = treatment replicates; a, b, c = different bottles.

Linalool (Figure 9) is one of the most volatile compounds in hops, regardless of the specific variety. Therefore, it is considered a fundamental compound (marker) in defining the aroma (hoppolate) of the finished beer [56,57]. It has been shown that linalool is also subject to additive and synergistic effects with other compounds, such as the by-products of fermentation, and that, consequently, it influences the perception of floral notes in beer. The threshold of perception of (R)-linalool is 2.2 µg/L [58,59].

The scent of geraniol (Figure 10) is a combination of rose, lime, and flowers. In contrast, the cis isomer (nerol) has a refreshing, "green" aroma [18]. During fermentation, the activity of acetate esterase can cause geranyl esters—commonly found in various hop varieties like Cascade—to be broken down into geraniol. Beers with a rich geraniol content are attained by using hop varieties containing high concentrations of geranyl esters. This suggests that these esters are hydrolysed during fermentation [1].

It has been observed how methyl esters (Figures 11 and 12) and derivatives of geraniol and linalool, such as linalool oxide and geranyl acetate, interact with each other to provide beer with fruity, green, floral aromatic notes but also waxy notes [60,61].

The sensory profiles of ketones (Figure 13) are significantly influenced by concentration and molecular weight. As the molecular weight increases, the fruity scent transitions into a floral fragrance. For instance, β-ionone and 2-undecanone (see Appendix A, Table 2) are known to impart floral aromatic notes [62], but also fruity [63] and citrus [53] aromatic notes at different strengths.

Although dip hopping (B beers) does not appear to have a negative impact on the fermentation activity of yeast (Table 5), it seems to affect the composition of secondary products, thus reducing the production of traditional acetic and ethyl esters compared with the reference beers (A). Specifically, isoamyl acetate (Figure 14), ethyl butanoate (Figure 15), hexanoate, and octanoate (Appendix A, Table 3) appear to prevail in A beers over B beers.

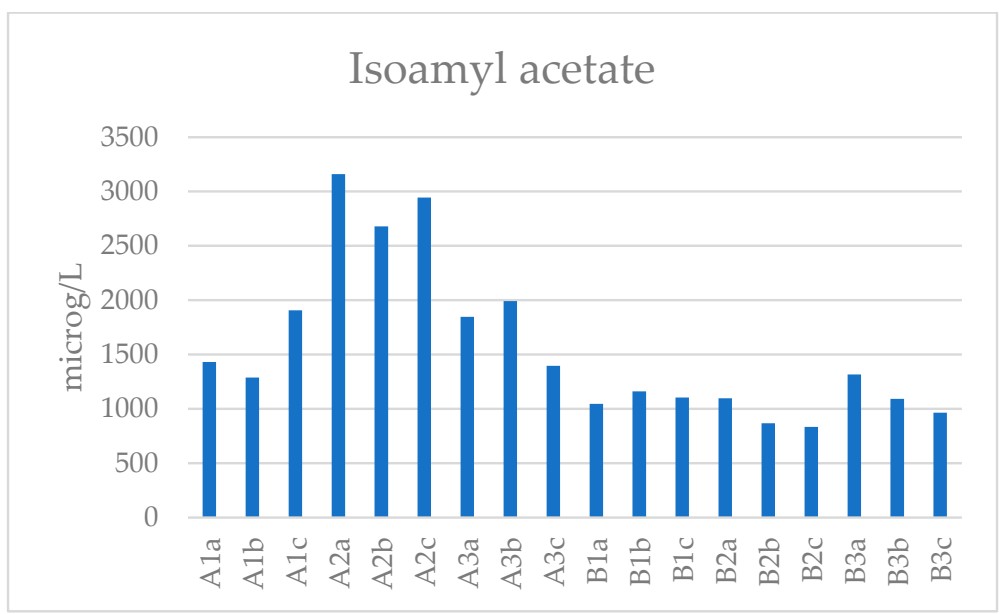

**Figure 14.** Concentration (SPME-GC-MS analysis) of isoamyl acetate in the late hopping (A) and dip hopping (B) beers. 1, 2, 3 = treatment replicates; a, b, c = different bottles.

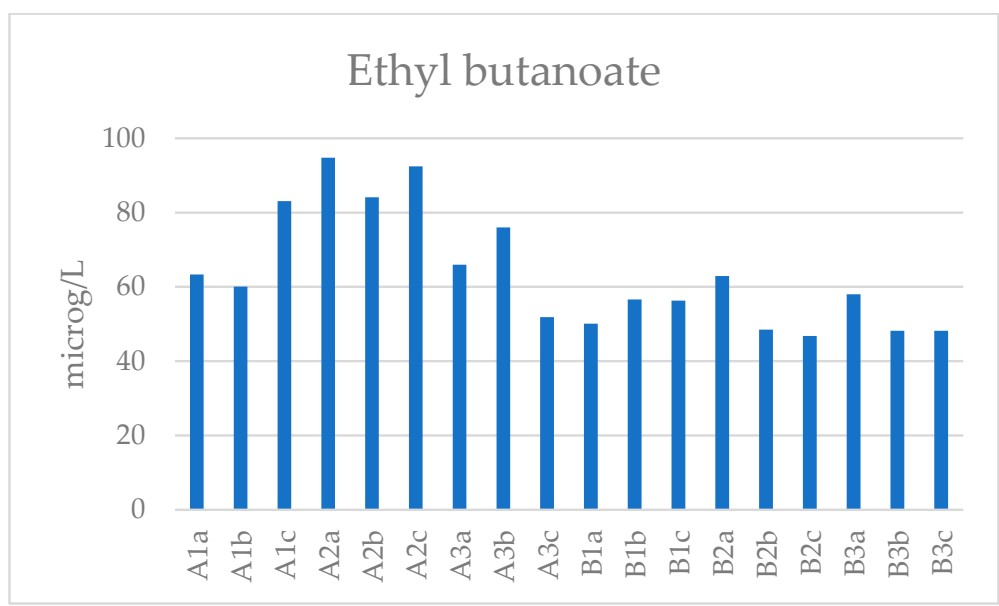

**Figure 15.** Concentration (SPME-GC-MS analysis) of ethyl butanoate in the late hopping (A) and dip hopping (B) beers. 1, 2, 3 = treatment replicates; a, b, c = different bottles.

The compounds were grouped into the following classes for statistical processing: terpenes, terpenols, alcohols, ketones, higher alcohol esters, acids, acetic esters, and ethyl esters. The box plots reflect the trends of the PCA (Figures 16–23). The results of the Kruskal-Wallis test show significant differences in all cases, while the more severe multiple comparison test also shows some significant differences. Terpenes (associated with citrus, fruity, spicy, and floral notes) and alcohols and ketones (associated with green and vegetable notes) are significantly higher in experimental beers (B) than in reference beers (A) (see Figures 16–19).

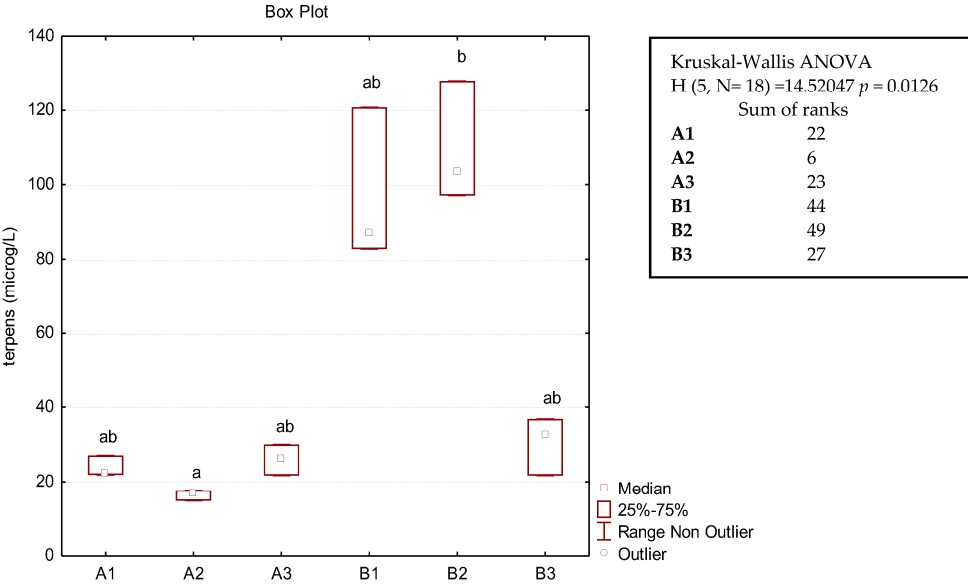

**Figure 16.** Box plot relating to terpenes. A and B are reference and experimental beers produced using Idaho hops; 1, 2, and 3 are treatment replicates. Statistically significant differences ($p \leq 0.05$) based on the multiple comparison test of the mean ranks for all groups are represented with different letters.

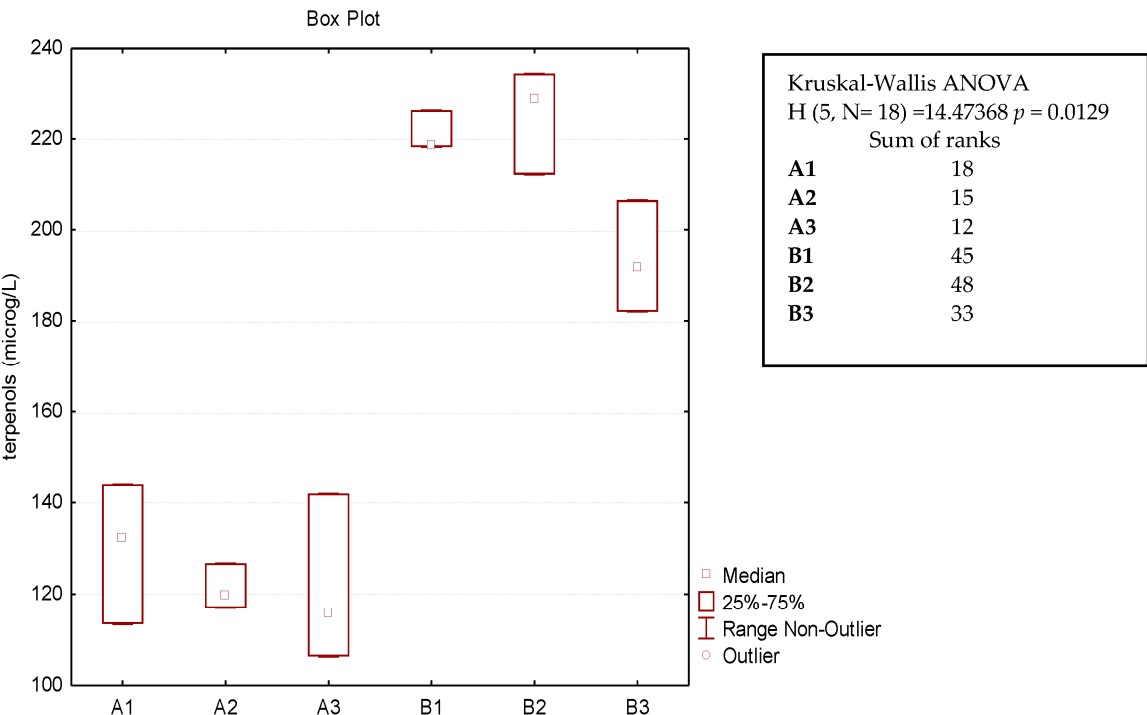

**Figure 17.** Box plot relating to terpenols. A and B = reference and experimental beers produced using Idaho hops; 1, 2, 3 = treatment replicates. Statistically significant differences ($p \leq 0.05$) based on the multiple comparison test of the mean ranks for all groups are represented with different letters.

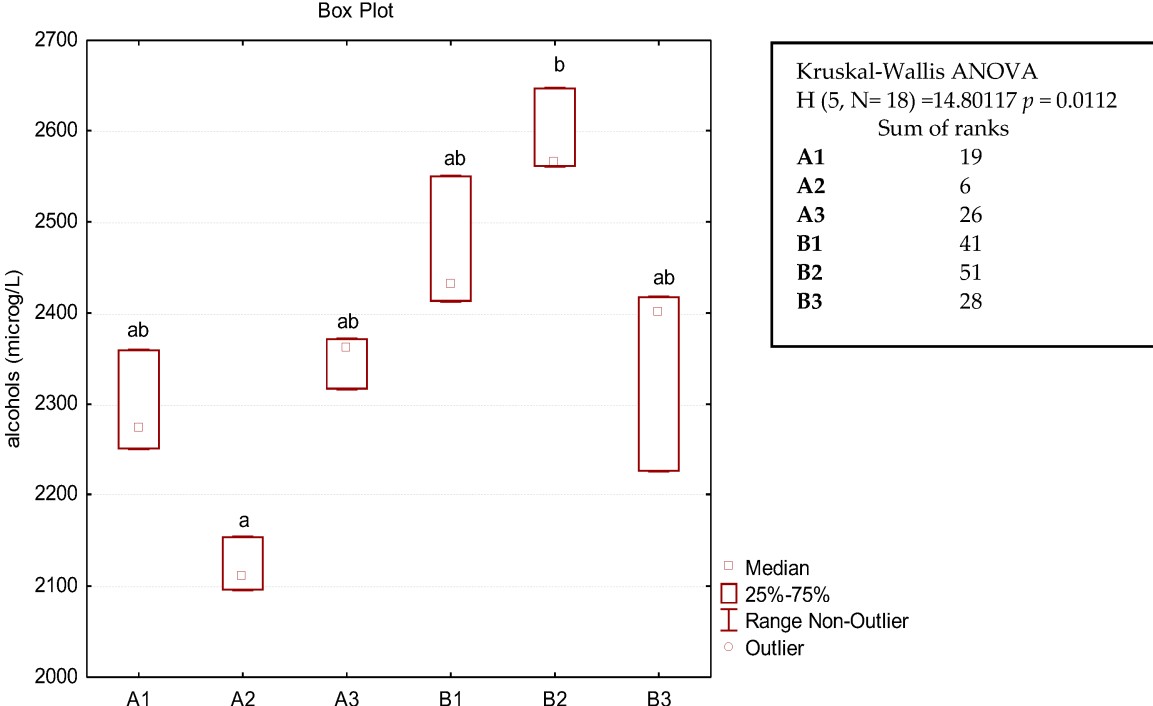

**Figure 18.** Box plot relating to alcohols. A and B = reference and experimental beers produced using Idaho hops; 1, 2, 3 = treatment replicates. Statistically significant differences ($p \leq 0.05$) based on the multiple comparison test of the mean ranks for all groups are represented with different letters.

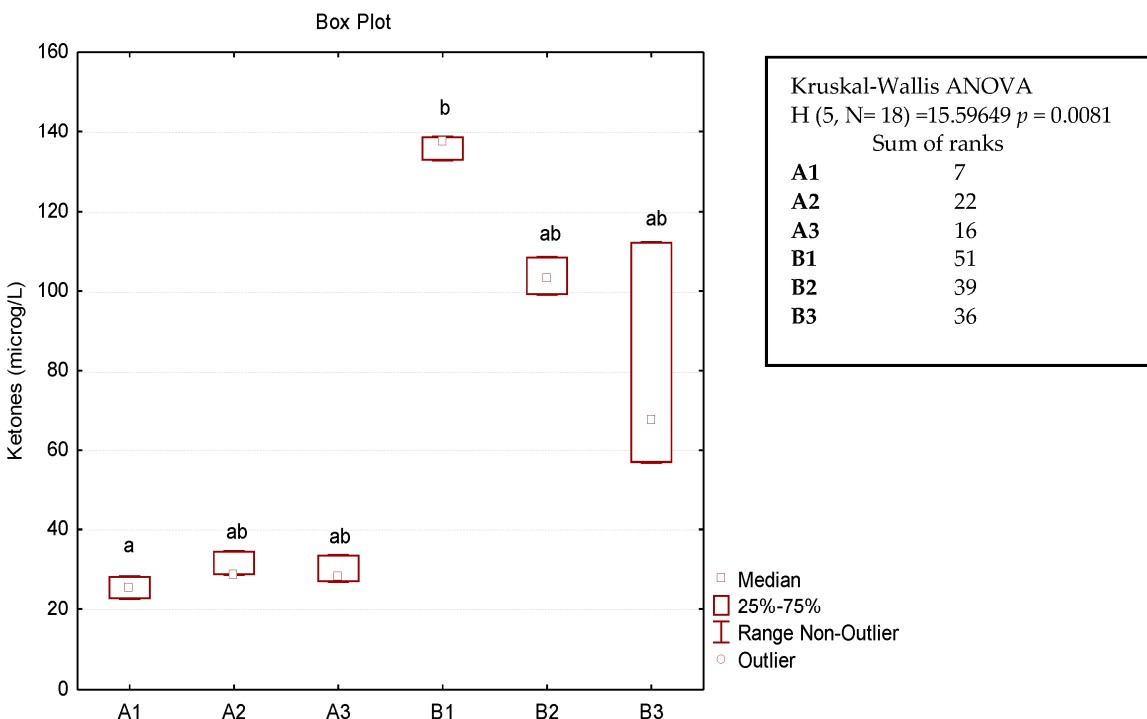

**Figure 19.** Box plot relating to ketones. A and B = reference and experimental beers produced using Idaho hops; 1, 2, and 3 = treatment replicates. Statistically significant differences ($p \leq 0.05$) based on the multiple comparison test of the mean ranks for all groups are represented with different letters.

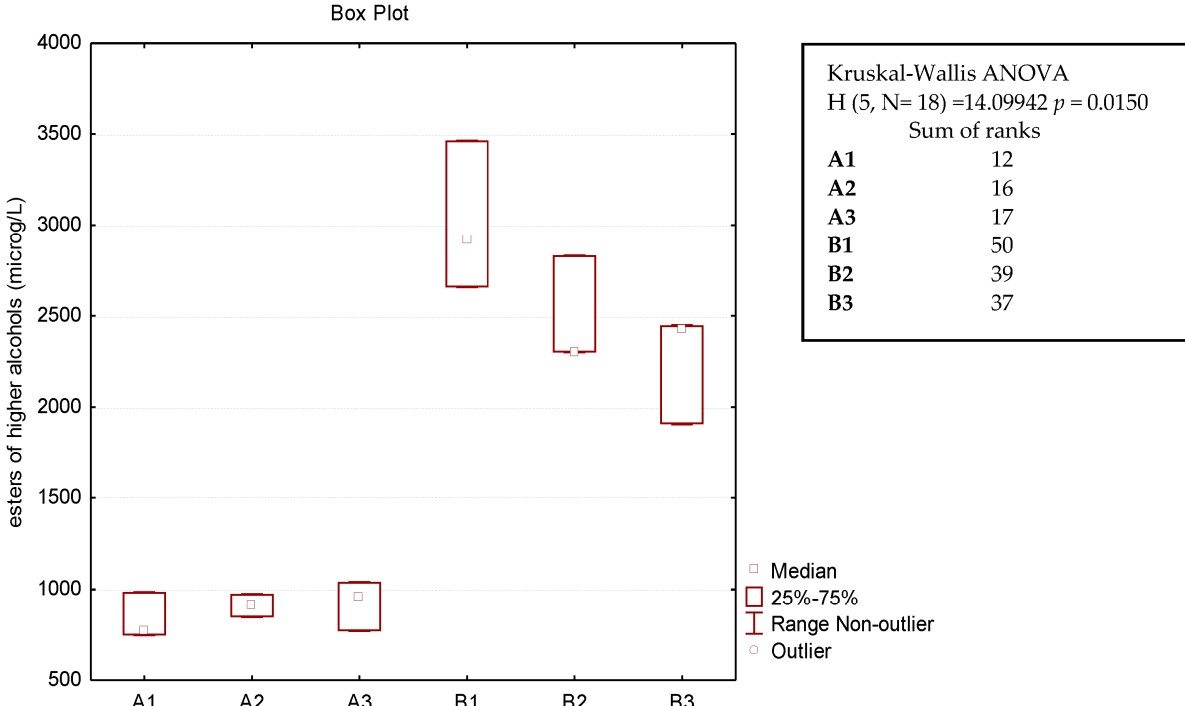

**Figure 20.** Box plot relating to higher alcohol esters. A and B are reference and experimental beers produced using Idaho hops; 1, 2, and 3 are treatment replicates. Statistically significant differences ($p \leq 0.05$) based on the multiple comparison test of the mean ranks for all groups are represented with different letters.

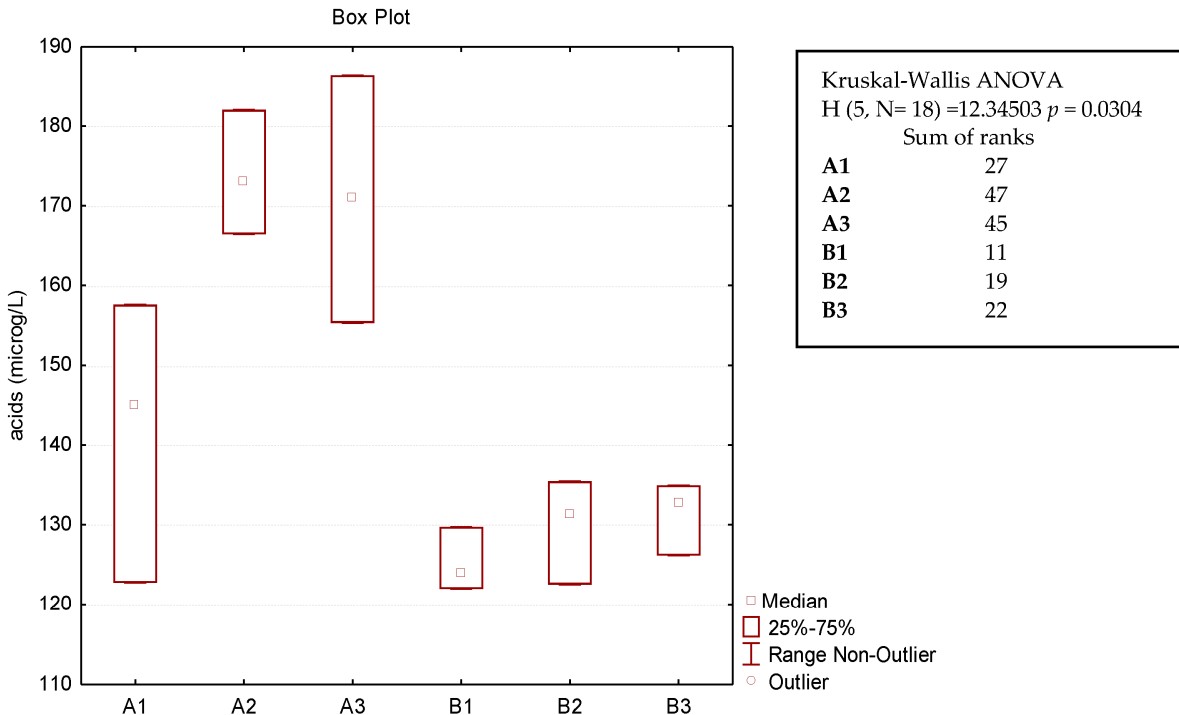

**Figure 21.** Box plot relating to acids. A and B are reference and experimental beers produced using Idaho hops; 1, 2, and 3 are treatment replicates. Statistically significant differences ($p \leq 0.05$) based on the multiple comparison test of the mean ranks for all groups are represented with different letters.

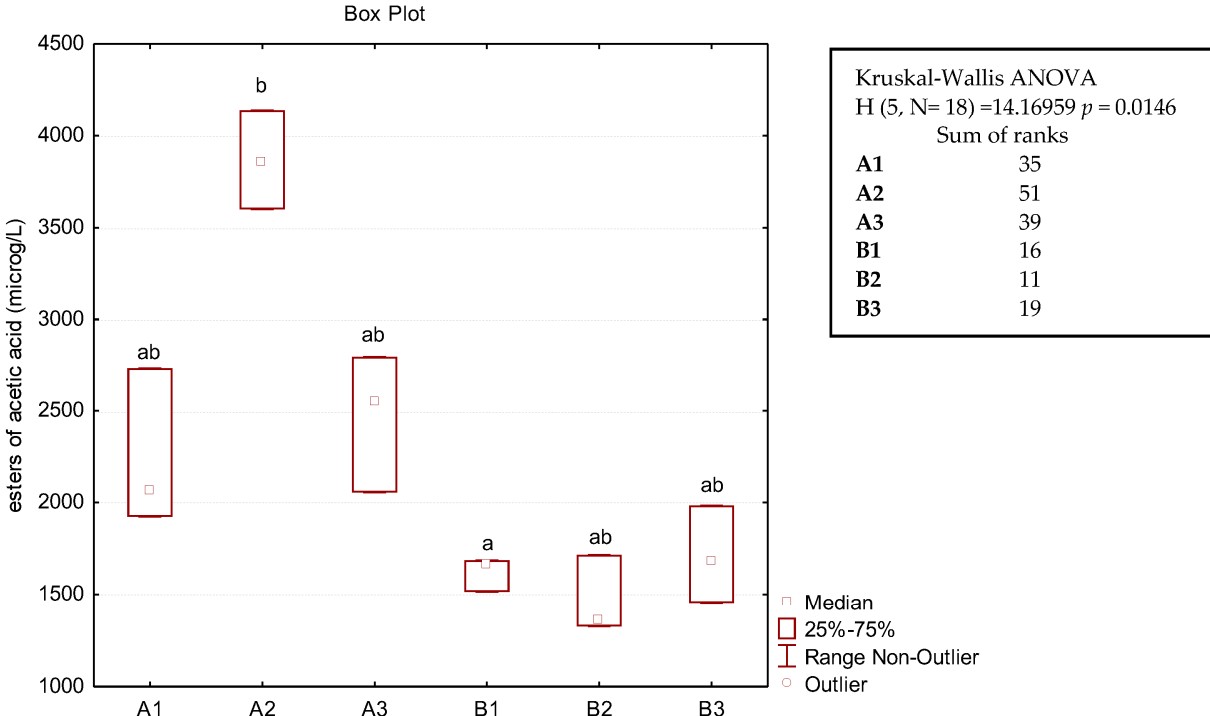

**Figure 22.** Box plot relating to acetic esters. A and B are reference and experimental beers produced using Idaho hops; 1, 2, and 3 are treatment replicates. Statistically significant differences ($p \leq 0.05$) based on the multiple comparison test of the mean ranks for all groups are represented with different letters.

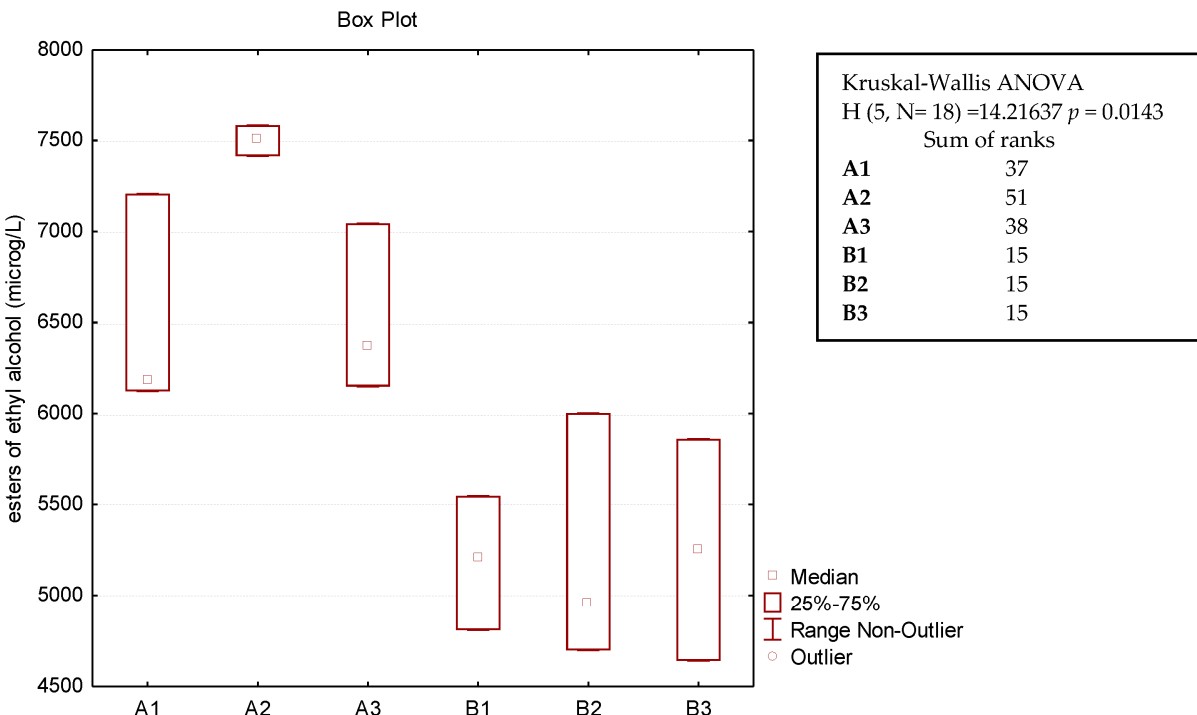

**Figure 23.** Box plot relating to ethyl esters. A and B = reference and experimental beers produced using Idaho hops; 1, 2, and 3 = treatment replicates. Statistically significant differences ($p \leq 0.05$) based on the multiple comparison test of the mean ranks for all groups are represented with different letters.

Acetic esters (associated with floral, fruity notes), which are closely related to alcoholic fermentation, are lower in the experimental beers (B) compared with the reference beers (A) (see Figure 22). Nonetheless, it should be borne in mind that the interactions involving the various compounds present in a complex mixture such as beer complicate the association of individual volatile substances with specific sensory perceptions.

The aroma of beer can be regulated by changing the hops' variety, adjusting the quantity and timing of their addition, and also by promoting specific yeast-mediated reactions during the fermentation process (biotransformations). Finally, exploiting biotransformations to enhance the flavour of the product can ultimately result in fewer hops being used, resulting in a more sustainable production process with reduced costs and environmental impact.

## 4. Conclusions

The results obtained have highlighted the potential of this novel hopping technique (dip hopping) as a viable way of reducing the use of hops, thus improving the sustainability of production. In fact, this technique generally accentuates the perception of positive descriptors such as floral, fruity, citrus, and spicy aromas while lessening negative oxidised, reductive, medicinal, and garlic notes. It can be assumed that the perception of said defects decreases as a result of the masking effect of the positive aromatic notes. The overall effect, in any event, remains an increase in the olfactory and gustatory pleasantness of beers produced employing the dip hopping technique. The addition of the infusion during the inoculation of a yeast with high enzymatic activity (beta-glucosidase) aided in the characterisation of the beer aroma. This is probably a direct result of the biotransformations by the yeast cells themselves on the molecules provided by the hops. The activity of the beta-glucosidase enzyme leads to the release of aromatic terpenes from non-aromatic glycosides. Furthermore, yeast cells interact chemically by hydrolysing esters, for example. Additive and synergistic effects also take place between its (yeast) by-products and hop

terpenes. Although dip hopping does not appear to negatively impact yeast fermentation, it does influence the composition of secondary products, leading to a reduction in the production of traditional acetic and ethyl esters. The results of the gas chromatography analyses also attest to this positive effect, indicating a prevalence of terpenes, terpenols, alcohols, ketones, and higher alcohol esters in beers brewed with the Idaho variety using the dip hopping technique. Further trials are ongoing to test the dip hopping technique on other hop varieties. The aim for the future is to identify to what extent the amount of hops can be reduced (with the dip hopping technique) while preserving olfactory characteristics that are comparable or superior to those of beers produced with the late hopping technique. Furthermore, dip hopping should also be compared to dry hopping, since in theory the former lies somewhere in between late hopping and dry techniques, generating more aromatic beers (compared with those produced employing late hopping) with a potentially more subdued hop flavour than the notes imparted by dry hopping. In the future, a comparison of dip hopping and dry hopping techniques in terms of the persistence of the aromatic notes over time could be of interest.

**Author Contributions:** P.P.: Writing—drawing up of the original draft; L.T. carried out the GC analyses and processed the data obtained. She processed the data obtained from the sensory analysis; A.G. carried out chemical analysis data and interpreted the results; L.V. carried out chemical analysis data and interpreted the results; S.B.: Writing—drawing up of the original draft and manuscript proofreading. All authors have read and agreed to the published version of the manuscript.

**Funding:** This research received no external funding.

**Institutional Review Board Statement:** The study was conducted in accordance with the Declaration of Helsinki, and approved by the Institutional Review Board of Agricultural, Food, Environmental and Animal Sciences, University of Udine (prot. N. 0006196, 28 November 2023).

**Informed Consent Statement:** Informed consent was obtained from all subjects involved in this study.

**Data Availability Statement:** Data are contained within the article.

**Acknowledgments:** No funding was received for conducting this study. We would like to thank Severino Garlatti Costa (president of the Associazione Artigiani Birrai [Association of Artisans Brewers] in Friuli-Venezia Giulia) and Manfredi Guglielmotti (technical consultant c/o P.A.B. SRL, Mr. Malt®, Pasian di Prato, Udine) for their contributions to creating the recipe. We would also like to thank Saida Favotto for organising the sensory analysis session.

**Conflicts of Interest:** The authors declare no conflicts of interest.

## Appendix A

**Table 1.** Volatile compounds (SPME-GC-MS analysis) in the late hopping (A) and dip hopping (B) beers. 1, 2, 3 = treatment replicates; a, b, c = different bottles.

| compound terpens | code | RT | Kovats' RI | A1a | A1b | A1c | A2a | A2b | A2c | A3a | A3b | A3c | B1a | B1b | B1c | B2a | B2b | B2c | B3a | B3b | B3c |
|---|---|---|---|---|---|---|---|---|---|---|---|---|---|---|---|---|---|---|---|---|---|
| | | | | | | | | | | | | μg/L | | | | | | | | | |
| alpha-myrcene | t1 | 10.51 | 1148 | 0.01 | 0.02 | 0.02 | 0.01 | 0.01 | 0.00 | 0.01 | 0.03 | 0.02 | 0.10 | 0.17 | 0.12 | 0.24 | 0.14 | 0.13 | 0.03 | 0.03 | 0.02 |
| alpha.-Phellandrene | t2 | 10.71 | 1154 | 0.16 | 0.21 | 0.14 | 0.09 | 0.12 | 0.10 | 0.14 | 0.25 | 0.22 | 0.26 | 0.35 | 0.34 | 0.47 | 0.32 | 0.33 | 0.24 | 0.26 | 0.21 |
| beta.-Myrcene | t3 | 10.91 | 1159 | 10.87 | 14.73 | 11.93 | 7.67 | 10.33 | 9.42 | 9.76 | 15.01 | 12.82 | 62.97 | 91.13 | 68.95 | 105.16 | 79.69 | 83.74 | 23.06 | 20.74 | 10.60 |
| D-Limonene | t4 | 12.05 | 1187 | 0.20 | 0.22 | 0.17 | 0.11 | 0.13 | 0.12 | 0.20 | 0.19 | 0.23 | 0.49 | 0.64 | 0.49 | 0.88 | 0.58 | 0.59 | 0.32 | 0.31 | 0.26 |
| cis-beta-Ocimene | t5 | 14.32 | 1249 | 0.54 | 0.54 | 0.41 | 0.29 | 0.27 | 0.28 | 0.38 | 3.50 | 0.43 | 2.19 | 3.35 | 2.43 | 4.29 | 2.66 | 3.08 | 0.75 | 0.70 | 0.52 |
| o-Cymene | t6 | 14.87 | 1264 | 7.55 | 8.26 | 7.45 | 5.07 | 5.14 | 5.23 | 9.43 | 8.11 | 9.11 | 8.04 | 7.93 | 8.26 | 9.40 | 8.15 | 8.18 | 9.67 | 8.62 | 7.69 |
| delta-Carene | t7 | 15.33 | 1275 | 0.04 | 0.06 | 0.04 | 0.02 | 0.04 | 0.03 | 0.04 | 0.05 | 0.07 | 0.12 | 0.16 | 0.14 | 0.23 | 0.18 | 0.16 | 0.09 | 0.10 | 0.09 |
| Caryophyllene | t9 | 25.78 | 1588 | 0.72 | 0.21 | 0.42 | 0.29 | 0.06 | 0.29 | 0.28 | 0.68 | 0.69 | 1.97 | 4.59 | 0.63 | 0.86 | 0.86 | 1.92 | 0.52 | 0.22 | 0.36 |
| Humulene | t11 | 27.93 | 1662 | 1.16 | 1.56 | 0.70 | 0.63 | 0.84 | 0.65 | 0.65 | 1.12 | 1.32 | 3.77 | 7.50 | 3.33 | 4.01 | 2.71 | 3.12 | 0.94 | 0.76 | 0.93 |
| gamma-Muurolene | t12 | 28.58 | 1683 | 0.22 | 0.22 | 0.10 | 0.09 | 0.11 | 0.07 | 0.19 | 0.20 | 0.23 | 0.48 | 0.77 | 0.40 | 0.43 | 0.29 | 0.30 | 0.26 | 0.17 | 0.14 |
| alpha-Terpineol | t14 | 29.00 | 1697 | 0.30 | 0.33 | 0.22 | 0.17 | 0.18 | 0.16 | 0.24 | 0.21 | 0.36 | 0.39 | 0.43 | 0.35 | 0.52 | 0.45 | 0.46 | 0.44 | 0.43 | 0.40 |
| Epizonarene | t15 | 29.24 | 1706 | 0.05 | 0.05 | 0.03 | 0.03 | 0.04 | 0.04 | 0.04 | 0.07 | 0.05 | 0.15 | 0.35 | 0.10 | 0.12 | 0.08 | 0.11 | 0.04 | 0.04 | 0.04 |
| beta-Selinene | t16 | 29.38 | 1711 | 0.05 | 0.08 | 0.02 | 0.05 | 0.05 | 0.04 | 0.05 | 0.06 | 0.07 | 0.19 | 0.41 | 0.13 | 0.16 | 0.11 | 0.14 | 0.05 | 0.06 | 0.05 |
| alpha-Selinene | t17 | 29.52 | 1716 | 0.02 | 0.04 | 0.01 | 0.01 | 0.02 | 0.01 | 0.02 | 0.02 | 0.03 | 0.08 | 0.16 | 0.07 | 0.07 | 0.01 | 0.01 | 0.02 | 0.01 | 0.02 |
| alpha-Muurolene | t18 | 29.55 | 1717 | 0.04 | 0.06 | 0.02 | 0.03 | 0.03 | 0.02 | 0.03 | 0.05 | 0.05 | 0.13 | 0.27 | 0.11 | 0.11 | 0.08 | 0.10 | 0.03 | 0.03 | 0.04 |
| delta-Cadinene | t20 | 30.53 | 1753 | 0.35 | 0.45 | 0.19 | 0.21 | 0.28 | 0.22 | 0.26 | 0.42 | 0.45 | 1.27 | 2.24 | 1.13 | 0.95 | 0.68 | 0.87 | 0.30 | 0.25 | 0.29 |
| alpha-Cadinene | t23 | 31.54 | 1787 | 0.02 | 0.03 | 0.01 | 0.01 | 0.01 | 0.01 | 0.01 | 0.02 | 0.02 | 0.06 | 0.11 | 0.05 | 0.05 | 0.04 | 0.04 | 0.02 | 0.02 | 0.02 |
| trans-Calamenene | t25 | 32.52 | 1825 | 0.04 | 0.04 | 0.02 | 0.03 | 0.03 | 0.02 | 0.03 | 0.04 | 0.05 | 0.09 | 0.14 | 0.08 | 0.10 | 0.07 | 0.09 | 0.04 | 0.05 | 0.04 |

| compound terpenols | code | RT | Kovats' RI | A1a | A1b | A1c | A2a | A2b | A2c | A3a | A3b | A3c | B1a | B1b | B1c | B2a | B2b | B2c | B3a | B3b | B3c |
|---|---|---|---|---|---|---|---|---|---|---|---|---|---|---|---|---|---|---|---|---|---|
| | | | | | | | | | | | | μg/L | | | | | | | | | |
| Terpinen-4-ol | t10 a | 26.19 | 1601 | 1.43 | 1.33 | 1.48 | 1.77 | 1.80 | 1.90 | 1.18 | 0.91 | 1.80 | 2.25 | 1.43 | 1.75 | 1.32 | 2.16 | 2.36 | 2.10 | 1.56 | 2.46 |
| Methyl geranate | t13 e | 28.84 | 1692 | 14.49 | 16.69 | 12.26 | 14.57 | 16.46 | 14.73 | 12.96 | 11.80 | 15.79 | 33.07 | 31.76 | 30.57 | 30.60 | 32.11 | 33.54 | 27.53 | 30.06 | 23.38 |
| NI * | t19 a | 30.30 | 1745 | 1.01 | 1.10 | 1.02 | 0.99 | 1.02 | 1.00 | 0.84 | 1.01 | 1.20 | 1.50 | 1.24 | 1.77 | 2.42 | 2.08 | 2.41 | 1.78 | 2.48 | 1.62 |
| alfa-Citronellol | t21 a | 30.81 | 1763 | 0.96 | 1.14 | 0.97 | 0.81 | 0.93 | 0.88 | 1.19 | 1.12 | 1.38 | 1.42 | 1.34 | 1.34 | 1.28 | 1.37 | 1.28 | 0.77 | 0.86 | 0.79 |
| beta-Citronellol | t22 a | 31.03 | 1767 | 33.34 | 38.01 | 27.58 | 24.54 | 29.69 | 25.93 | 33.31 | 29.25 | 41.13 | 46.22 | 44.35 | 42.48 | 40.98 | 44.59 | 45.53 | 25.46 | 29.39 | 26.08 |
| Nerol | t24 a | 31.94 | 1803 | 1.45 | 1.62 | 1.14 | 1.36 | 1.44 | 1.38 | 1.07 | 1.16 | 1.69 | 2.41 | 2.12 | 2.35 | 2.42 | 2.40 | 2.51 | 2.01 | 2.42 | 2.20 |
| Geraniol | t26 a | 33.21 | 1852 | 3.94 | 4.47 | 2.93 | 2.42 | 2.77 | 3.17 | 2.03 | 1.85 | 3.00 | 7.91 | 7.83 | 7.94 | 5.58 | 6.94 | 7.14 | 7.65 | 8.79 | 7.47 |
| Linalool | t8 a | 24.62 | 1551 | 75.57 | 79.81 | 66.00 | 70.51 | 72.68 | 70.87 | 63.20 | 59.18 | 75.99 | 131.71 | 128.48 | 130.01 | 127.56 | 137.46 | 139.72 | 124.34 | 130.94 | 118.03 |
| β-Damascenone | nor | 32.32 | 1817 | 0.83 | 0.83 | 0.66 | 0.45 | 0.83 | 0.67 | 0.56 | 0.56 | 1.07 | 0.88 | 0.84 | 0.76 | 0.87 | 0.86 | 0.84 | 1.27 | 1.36 | 1.68 |

* = Not Identified

**Table 2.** Volatile compounds (SPME-GC-MS analysis) in the late hopping (A) and dip hopping (B) beers. 1, 2, 3 = treatment replicates; a, b, c = different bottles.

| compound acids | code | RT | Kovats' RI | A1a | A1b | A1c | A2a | A2b | A2c | A3a | μg/L A3b | A3c | B1a | B1b | B1c | B2a | B2b | B2c | B3a | B3b | B3c |
|---|---|---|---|---|---|---|---|---|---|---|---|---|---|---|---|---|---|---|---|---|---|
| Acetic acid | ac1 | 21.38 | 1450 | 7.94 | 8.40 | 8.25 | 25.65 | 25.24 | 28.43 | 45.24 | 42.68 | 38.92 | 6.47 | 5.34 | 6.92 | 14.94 | 13.69 | 13.10 | 3.95 | 10.17 | 7.23 |
| Propanoic acid, 2-methyl- | ac2 | 25.16 | 1569 | 16.41 | 17.86 | 18.05 | 16.78 | 17.39 | 17.60 | 19.06 | 20.86 | 19.33 | 23.71 | 25.89 | 24.34 | 24.63 | 24.46 | 24.84 | 21.82 | 20.54 | 19.07 |
| Butanoic acid | ac3 | 26.97 | 1628 | 2.54 | 2.72 | 2.60 | 3.18 | 2.84 | 3.12 | 3.76 | 2.91 | 2.43 | 2.88 | 2.42 | 2.30 | 2.39 | 2.14 | 2.12 | 4.73 | 2.20 | 2.01 |
| Butanoic acid, 2 and 3-methyl- | ac4 | 28.21 | 1671 | 19.26 | 19.02 | 18.57 | 18.98 | 18.61 | 19.16 | 20.94 | 20.26 | 19.92 | 19.40 | 19.68 | 19.50 | 22.00 | 21.66 | 21.78 | 19.87 | 19.28 | 19.14 |
| Heptanoic acid | ac5 | 35.89 | 1954 | 2.75 | 3.04 | 2.08 | 2.61 | 2.72 | 2.46 | 2.61 | 2.14 | 2.42 | 3.58 | 3.36 | 3.29 | 3.70 | 4.45 | 4.46 | 4.33 | 4.10 | 3.86 |
| 6-Methylheptanoic acid | ac6 | 37.36 | 2012 | 2.34 | 2.63 | 1.94 | 2.07 | 2.03 | 1.72 | 2.20 | 1.63 | 2.56 | 3.72 | 3.58 | 3.28 | 3.32 | 3.80 | 3.66 | 4.04 | 3.54 | 3.57 |
| Octanoic acid | ac7 | 38.55 | 2062 | 78.59 | 87.64 | 62.71 | 98.33 | 85.90 | 88.69 | 66.67 | 55.50 | 86.15 | 57.26 | 52.37 | 53.43 | 43.54 | 53.83 | 51.12 | 62.73 | 56.64 | 62.62 |
| Nonanoic acid | ac8 | 41.09 | 2169 | 1.39 | 2.14 | 1.30 | 1.64 | 1.52 | 1.51 | 1.95 | 1.21 | 2.30 | 2.52 | 1.98 | 1.93 | 1.93 | 2.60 | 2.38 | 4.04 | 2.23 | 1.90 |
| n-Decanoic acid | ac9 | 43.51 | 2276 | 8.10 | 7.82 | 4.40 | 8.14 | 6.03 | 5.84 | 4.68 | 4.48 | 6.67 | 3.80 | 3.31 | 3.86 | 2.28 | 3.27 | 2.76 | 4.79 | 1.94 | 6.68 |
| Dodecanoic acid | ac10 | 48.06 | 2489 | 5.73 | 6.32 | 2.84 | 4.77 | 4.14 | 4.47 | 3.93 | 3.65 | 5.78 | 6.38 | 4.04 | 5.18 | 3.77 | 5.54 | 5.19 | 4.63 | 5.53 | 6.70 |
| **alchools** | | | | | | | | | | | | | | | | | | | | | |
| 1-Propanol, 2-methyl- | al1 | 9.07 | 1107 | 153.62 | 146.48 | 168.78 | 131.12 | 133.70 | 137.30 | 192.85 | 193.98 | 172.15 | 168.87 | 176.91 | 178.48 | 197.13 | 189.57 | 206.36 | 166.62 | 157.37 | 136.87 |
| 1-Butanol, 3-methyl- | al2 | 13.24 | 1218 | 1337.41 | 1330.01 | 1537.25 | 1304.30 | 1306.73 | 1352.17 | 1478.33 | 1523.26 | 1327.35 | 1308.13 | 1339.27 | 1450.38 | 1488.33 | 1410.87 | 1486.49 | 1399.52 | 1337.38 | 1251.24 |
| 1-Hexanol | al3 | 18.32 | 1360 | 9.88 | 9.83 | 10.46 | 13.07 | 3.00 | 13.80 | 9.98 | 10.27 | 12.10 | 13.78 | 13.58 | 14.71 | 15.46 | 22.14 | 14.98 | 15.15 | 16.04 | 20.50 |
| 1-Octen-3-ol | al4 | 21.56 | 1455 | 3.96 | 3.88 | 3.15 | 4.21 | 4.22 | 4.11 | 2.94 | 3.16 | 3.07 | 14.37 | 11.86 | 13.74 | 7.64 | 7.14 | 6.95 | 9.43 | 9.85 | 9.58 |
| 2-Nonanol | al5 | 23.83 | 1525 | 74.67 | 77.99 | 67.64 | 62.31 | 67.32 | 64.42 | 70.91 | 66.88 | 77.41 | 151.15 | 146.35 | 146.06 | 146.42 | 156.37 | 156.72 | 150.81 | 152.14 | 142.18 |
| 1-Octanol | al6 | 24.99 | 1563 | 15.41 | 15.73 | 13.46 | 19.08 | 18.44 | 18.67 | 12.02 | 11.80 | 14.52 | 13.63 | 13.38 | 14.72 | 13.25 | 14.88 | 14.01 | 16.22 | 16.99 | 15.26 |
| NI * | al7 | 25.52 | 1580 | 7.25 | 7.39 | 6.88 | 4.63 | 5.21 | 4.47 | 6.31 | 5.74 | 8.07 | 15.31 | 14.46 | 14.54 | 15.54 | 16.63 | 17.16 | 15.86 | 14.55 | 13.13 |

| compound acohols | code | RT | Kovats' RI | A1a | A1b | A1c | A2a | A2b | A2c | A3a | μg/L A3b | A3c | B1a | B1b | B1c | B2a | B2b | B2c | B3a | B3b | B3c |
|---|---|---|---|---|---|---|---|---|---|---|---|---|---|---|---|---|---|---|---|---|---|
| 2-Decanol | al8 | 26.87 | 1625 | 24.21 | 26.22 | 21.42 | 20.40 | 23.90 | 21.65 | 22.49 | 21.17 | 27.09 | 57.45 | 53.37 | 54.27 | 55.05 | 61.82 | 63.16 | 56.40 | 63.76 | 51.42 |
| NI* | al9 | 28.45 | 1679 | 5.67 | 6.23 | 4.95 | 7.65 | 8.15 | 7.89 | 5.19 | 4.73 | 6.26 | 10.29 | 11.04 | 11.71 | 11.79 | 13.56 | 11.93 | 10.59 | 12.33 | 7.81 |
| 2-Undecanol | al10 | 29.76 | 1725 | 48.01 | 53.43 | 42.01 | 42.86 | 52.67 | 45.77 | 42.36 | 39.40 | 53.54 | 105.74 | 95.38 | 100.92 | 92.64 | 106.15 | 111.15 | 80.48 | 93.99 | 73.42 |
| 1-Decanol | al11 | 30.94 | 1768 | 4.88 | 5.27 | 4.36 | 3.50 | 3.87 | 3.47 | 2.96 | 3.30 | 4.02 | 3.37 | 3.36 | 3.70 | 2.44 | 3.16 | 2.56 | 3.24 | 3.51 | 3.35 |
| Phenylethyl Alcohol | al12 | 34.79 | 1911 | 565.63 | 591.33 | 479.41 | 482.50 | 484.34 | 480.56 | 525.85 | 478.63 | 610.55 | 570.14 | 533.40 | 547.36 | 514.58 | 564.45 | 556.60 | 493.89 | 523.48 | 501.28 |
| **ketones** | | | | | | | | | | | | | | | | | | | | | |
| 2-nonanone | k1 | 19.27 | 1386 | 12.96 | 14.12 | 12.17 | 15.53 | 17.33 | 15.35 | 16.44 | 15.31 | 18.16 | 55.04 | 56.56 | 54.29 | 44.05 | 46.22 | 47.09 | 36.33 | 52.67 | 33.46 |
| NI * | k2 | 21.26 | 1446 | 0.77 | 0.82 | 0.70 | 0.81 | 1.06 | 0.88 | 1.05 | 0.92 | 1.13 | 7.61 | 7.50 | 7.47 | 5.65 | 6.05 | 6.26 | 3.93 | 6.90 | 3.33 |
| 2-Decanone | k3 | 22.76 | 1490 | 3.26 | 3.81 | 3.12 | 3.75 | 4.60 | 3.88 | 3.85 | 3.54 | 4.70 | 18.15 | 18.66 | 17.63 | 13.62 | 14.23 | 14.80 | 10.75 | 18.59 | 8.79 |
| 2-Undecanone | k4 | 26.03 | 1596 | 6.73 | 7.62 | 5.53 | 6.96 | 9.63 | 7.10 | 6.14 | 5.82 | 7.77 | 46.76 | 44.19 | 43.41 | 28.89 | 29.72 | 32.29 | 14.81 | 30.57 | 9.77 |
| 2-Dodecanone | k5 | 29.13 | 1702 | 0.46 | 0.54 | 0.40 | 0.46 | 0.57 | 0.44 | 0.31 | 0.35 | 0.46 | 2.96 | 2.77 | 2.72 | 1.96 | 2.02 | 2.30 | 0.62 | 1.30 | 0.49 |
| 2-Tridecanone | k6 | 32.06 | 1807 | 1.15 | 1.46 | 0.74 | 1.17 | 1.47 | 1.15 | 0.69 | 0.96 | 1.49 | 8.26 | 7.77 | 7.23 | 4.85 | 5.06 | 5.82 | 1.22 | 2.52 | 1.06 |
| NI * | NI1 | 29.94 | 1732 | 19.42 | 22.31 | 16.96 | 17.13 | 20.21 | 17.60 | 18.29 | 17.35 | 23.15 | 34.64 | 32.27 | 33.72 | 31.66 | 34.13 | 35.41 | 31.68 | 34.49 | 31.44 |
| NI * | NI2 | 30.41 | 1749 | 10.06 | 11.14 | 8.11 | 8.78 | 10.19 | 8.67 | 8.60 | 7.94 | 11.50 | 25.24 | 23.62 | 24.23 | 21.36 | 24.87 | 26.02 | 21.76 | 25.63 | 20.50 |

* = Not Identified

**Table 3.** Volatile compounds (SPME-GC-MS analysis) in the late hopping (A) and dip hopping (B) beers. 1, 2, 3 = treatment replicates; a, b, c = different bottles.

| compound esters | code | RT | Kovats' RI | A1a | A1b | A1c | A2a | A2b | A2c | A3a | µg/L A3b | A3c | B1a | B1b | B1c | B2a | B2b | B2c | B3a | B3b | B3c |
|---|---|---|---|---|---|---|---|---|---|---|---|---|---|---|---|---|---|---|---|---|---|
| Ethyl Acetate | ea1 | 3.17 | 892 | 511.10 | 475.40 | 670.31 | 602.39 | 606.76 | 627.25 | 543.26 | 631.76 | 467.32 | 356.62 | 404.17 | 412.10 | 481.62 | 367.33 | 382.80 | 464.22 | 402.07 | 365.26 |
| Isobutyl acetate | ea2 | 5.87 | 1016 | 28.11 | 50.74 | 55.04 | 120.94 | 97.94 | 53.93 | 46.27 | 67.38 | 56.28 | 22.15 | 28.09 | 62.03 | 54.95 | 35.67 | 30.95 | 98.81 | 96.24 | 35.36 |
| 1-Butanol, 3-methyl-, acetate | ea3 | 9.55 | 1121 | 1430.35 | 1288.18 | 1907.11 | 3158.90 | 2678.80 | 2944.36 | 1846.56 | 1992.10 | 1396.39 | 1046.35 | 1160.98 | 1103.65 | 1096.36 | 867.23 | 834.47 | 1316.70 | 1091.03 | 964.12 |
| Acetic acid, 2-phenylethyl ester | ea4 | 32.16 | 1811 | 100.51 | 109.14 | 97.74 | 260.72 | 216.81 | 226.37 | 113.58 | 106.19 | 134.87 | 89.66 | 93.78 | 87.52 | 82.62 | 85.10 | 75.99 | 104.25 | 91.44 | 89.64 |
| Butanoic acid, ethyl ester | ee1 | 6.61 | 1040 | 63.34 | 60.08 | 83.08 | 94.79 | 84.11 | 92.43 | 65.97 | 76.00 | 51.87 | 50.07 | 56.60 | 56.31 | 62.91 | 48.51 | 46.79 | 58.03 | 48.20 | 48.19 |
| Butanoic acid, 2-methyl-, ethyl ester | ee2 | 7.14 | 1056 | 12.90 | 11.87 | 16.95 | 9.68 | 10.82 | 10.52 | 17.66 | 17.81 | 12.82 | 15.61 | 16.76 | 18.70 | 25.48 | 17.75 | 19.46 | 19.91 | 18.20 | 16.26 |
| Hexanoic acid, ethyl ester | ee3 | 13.84 | 1236 | 824.66 | 829.95 | 1011.39 | 1355.32 | 1191.17 | 1355.99 | 935.07 | 983.43 | 833.44 | 697.17 | 811.06 | 806.96 | 894.99 | 750.14 | 676.32 | 795.37 | 691.95 | 746.69 |
| Ethyl 5-methylhexanoate | ee4 | 15.77 | 1286 | 45.42 | 36.83 | 42.92 | 26.58 | 30.59 | 32.89 | 39.57 | 35.41 | 35.17 | 52.32 | 61.81 | 75.04 | 76.56 | 66.62 | 60.86 | 60.80 | 60.05 | 56.28 |
| Heptanoic acid, ethyl ester | ee5 | 17.43 | 1334 | 196.54 | 190.14 | 230.16 | 180.45 | 222.04 | 190.61 | 251.39 | 285.53 | 213.92 | 313.61 | 358.38 | 315.43 | 452.48 | 341.42 | 359.31 | 310.47 | 339.47 | 293.30 |
| NI * | ee6 | 19.28 | 1386 | 75.65 | 75.63 | 89.55 | 65.42 | 77.40 | 70.93 | 83.03 | 95.41 | 77.48 | 122.69 | 143.97 | 129.73 | 178.98 | 141.07 | 135.18 | 141.56 | 123.41 | 116.19 |
| Octanoic acid, ethyl ester | ee7 | 21.06 | 1440 | 4044.21 | 4105.60 | 4832.94 | 4876.89 | 5016.11 | 4852.70 | 4203.34 | 4715.32 | 4060.80 | 2940.21 | 3386.47 | 3159.01 | 3657.51 | 2994.13 | 2834.60 | 3631.97 | 2860.81 | 3141.57 |
| Nonanoic acid, ethyl ester | ee8 | 24.18 | 1537 | 37.80 | 62.18 | 68.00 | 60.52 | 61.65 | 53.13 | 62.82 | 43.34 | 77.41 | 50.43 | 56.77 | 52.97 | 70.72 | 76.79 | 69.31 | 90.35 | 50.93 | 53.71 |
| Decanoic acid, ethyl ester | ee9 | 27.30 | 1640 | 688.25 | 638.26 | 683.32 | 712.07 | 658.82 | 604.31 | 517.59 | 640.50 | 553.77 | 341.49 | 407.35 | 396.71 | 337.25 | 309.06 | 291.41 | 405.15 | 173.54 | 544.58 |

| compound esters | code | RT | Kovats' RI | A1a | A1b | A1c | A2a | A2b | A2c | A3a | µg/L A3b | A3c | B1a | B1b | B1c | B2a | B2b | B2c | B3a | B3b | B3c |
|---|---|---|---|---|---|---|---|---|---|---|---|---|---|---|---|---|---|---|---|---|---|
| Ethyl trans-4-decenoate | ee10 | 28.08 | 1667 | 78.62 | 81.79 | 80.50 | 58.36 | 92.34 | 67.03 | 72.88 | 81.33 | 84.97 | 144.72 | 160.06 | 153.69 | 178.34 | 147.02 | 147.64 | 150.12 | 156.36 | 127.17 |
| Ethyl 9-decenoate | ee11 | 28.81 | 1691 | 56.28 | 91.96 | 67.75 | 69.43 | 139.08 | 85.07 | 125.24 | 70.93 | 147.66 | 78.88 | 88.20 | 46.99 | 68.06 | 66.78 | 56.90 | 196.21 | 118.03 | 105.12 |
| Propanoic acid, 2-methyl-, propyl ester | ae1 | 7.19 | 1057 | 18.13 | 18.91 | 26.04 | 15.98 | 20.70 | 18.27 | 22.40 | 23.29 | 16.34 | 41.58 | 45.64 | 45.09 | 42.83 | 32.07 | 30.24 | 36.97 | 36.35 | 29.02 |
| Propanoic acid, 2-methyl-, 2-methylpropyl ester | ae2 | 8.40 | 1089 | 244.98 | 237.56 | 341.72 | 247.21 | 287.95 | 266.99 | 292.61 | 325.06 | 224.65 | 674.12 | 805.10 | 747.80 | 789.99 | 612.36 | 605.80 | 688.28 | 661.96 | 542.11 |
| Propyl 2-methylbutyrate | ae3 | 10.08 | 1136 | 4.95 | 5.13 | 5.67 | 4.82 | 5.79 | 5.91 | 6.78 | 7.09 | 5.15 | 13.49 | 14.97 | 13.76 | 15.10 | 11.74 | 11.64 | 14.69 | 13.16 | 10.61 |
| Butanoic acid, 2-methyl-, 2-methylpropyl ester | ae4 | 11.55 | 1175 | 27.44 | 25.28 | 34.89 | 28.51 | 34.12 | 32.35 | 33.58 | 36.51 | 27.49 | 87.55 | 414.30 | 89.28 | 103.07 | 83.56 | 79.88 | 103.10 | 105.18 | 84.77 |
| Isobutyl isovalerate | ae5 | 12.14 | 1189 | 6.72 | 6.39 | 8.36 | 6.81 | 7.97 | 7.66 | 9.44 | 8.64 | 6.36 | 26.50 | 27.10 | 22.89 | 28.04 | 22.28 | 20.74 | 21.97 | 21.08 | 16.33 |
| Propanoic acid, 2-methyl-, 2 e 3-methylbutyl ester | ae6 | 12.33 | 1193 | 314.12 | 304.25 | 385.93 | 386.68 | 448.85 | 425.12 | 412.41 | 451.83 | 335.77 | 1436.15 | 1714.00 | 1572.66 | 1382.18 | 1135.34 | 1166.74 | 1196.66 | 1210.92 | 907.93 |
| Butanoic acid, 2-methyl-, 2-methylbutyl ester | ae7 | 15.48 | 1279 | 27.46 | 27.07 | 31.34 | 25.02 | 31.52 | 29.92 | 29.24 | 31.77 | 27.86 | 76.86 | 91.55 | 91.49 | 116.58 | 94.84 | 91.17 | 79.33 | 80.37 | 58.12 |
| Butanoic acid, 3-methyl-, 2-methylbutyl ester | ae8 | 16.08 | 1294 | 14.79 | 15.30 | 17.35 | 15.58 | 20.18 | 19.26 | 15.81 | 17.22 | 16.22 | 55.43 | 62.42 | 64.57 | 73.41 | 66.99 | 57.59 | 48.70 | 50.58 | 39.26 |
| NI * | ae9 | 16.78 | 1314 | 51.87 | 44.77 | 64.95 | 58.74 | 53.72 | 52.12 | 67.21 | 72.01 | 48.84 | 117.34 | 140.56 | 123.54 | 118.63 | 97.88 | 111.49 | 110.55 | 98.95 | 88.23 |
| Hexanoic acid, 4-methylene-, methyl ester | ae10 | 17.34 | 1331 | 62.29 | 60.22 | 71.35 | 55.19 | 64.72 | 59.40 | 64.11 | 67.63 | 60.65 | 127.62 | 149.39 | 144.64 | 162.23 | 142.07 | 127.27 | 148.06 | 142.81 | 129.93 |

\* = Not Identified

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
