# Peer review of "Dip Hopping Technique and Yeast Biotransformations in Craft Beer Productions"

_fermentation, doi:10.3390/fermentation10010030_

Round 1

Reviewer 1 Report

Comments and Suggestions for Authors

Reviewer’s comments

The aim of this paper is to evaluate the effects of an alternative hopping technique, called dip hopping, on beer production. This technique is an innovative, recently patented beer production method which gives beer with a better aroma and less hops. These are important studies confirming the validity of implementing this method. However, there are some comments regarding the submitted manuscript:

Minor comments

Line 82: Please change the word alpha into symbol α. The introduction used a symbol, not the whole word.

Line 104: The number 104 is in the wrong place

Table 2 and 3: Please rearrange both tables (2 and 3) as they are completely unreadable in this form. In the amount of grams, the number g is missing at the number 13. Additionally, the tables are incorrectly formatted. This needs to be improved. Table 3 contains only one column, so is it justified to present this data in the form of the table that is difficult to read?

Line 158: Please expand the title to what methods you mean. Using only the methods is not sufficient in this case.

Lines 192-194: The numbers are in the wrong places.

Lines 196 and 260: This point 2.3.1.2. is wrong. Please add to Statistical analysis point expanding what this analysis is about. There are two points with the same name, so they need to be clarified.

Figures 5 and 6: the description on these figures is completely illegible. The subtitles overlap.

Reviewer 2 Report

Comments and Suggestions for Authors

Upon reading the manuscript, my opinion is that the manuscript should be accepted with minor corrections.

Introduction:

Please explain the novelty and importance of the study more clearly.

Materials and methods:

The experimental design is completely unclear to the reader.

Please add a section of experimental design

The formulations should be described in more detail, including the name and location of ingredient suppliers.

“The sensory study plan was executed using a Counterbalance Design? What was the sensory plan?

Why were demographic traits not described in the methods? Please add a table showing the demographic characteristics of test participants. It is important to know how often sensory test participants used.

The research methodology is presented and the article is well laid out and structured to illustrate the unique properties of beer. However, the results of the paper are imperfect, and what advantages its unique attributes have, for example, how volatiles affect key product attributes are not specifically written, and it is recommended that the flavor characteristics of volatiles be added.

Results and discussion.

Please improved the discussion explaining the implication of the results.
